# Multi-Chain Graphs of Graphs: A New Approach to Analyzing Blockchain Datasets

**Bingqiao Luo**
National University of Singapore
luo.bingqiao@u.nus.edu

**Zhen Zhang**[*]
National University of Singapore
zhen@nus.edu.sg

**Qian Wang**
National University of Singapore
qiansoc@nus.edu.sg

**Bingsheng He**
National University of Singapore
hebs@comp.nus.edu.sg

## Abstract

Machine learning applied to blockchain graphs offers significant opportunities for enhanced data analysis and applications. However, the potential of this field is constrained by the lack of a large-scale, cross-chain dataset that includes hierarchical graph-level data. To address this issue, we present novel datasets that provide detailed label information at the token level and integrate interactions between tokens across multiple blockchain platforms. We model transactions within each token as local graphs and the relationships between tokens as global graphs, collectively forming a "Graphs of Graphs" (GoG) approach. This innovative approach facilitates a deeper understanding of systemic structures and hierarchical interactions, which are essential for applications such as link prediction, anomaly detection, and token classification. We conduct a series of experiments demonstrating that this dataset delivers new insights and challenges for exploring GoG within the blockchain domain. Our work promotes advancements and opens new avenues for research in both the blockchain and graph communities. Source code and datasets are available at https://github.com/Xtra-Computing/Cryptocurrency-Graphs-of-graphs.

## 1 Introduction

Machine learning techniques applied to blockchain graphs present significant opportunities for in-depth data analysis and innovative applications [1, 2]. A comprehensive analysis of graph structures and patterns reveals valuable insights into transaction activities, including the investigation of transaction patterns, identification of key players, and deployment of tokenomics frameworks [3, 2, 4, 5]. Advanced graph learning algorithms have shown promise in enhancing the detection of various fraudulent activities [6, 7, 8, 9, 10] and predicting market trends [11, 12]. However, existing studies in this field face significant limitations due to the restricted availability of open and extensive datasets that include graph-level data. Most labeled blockchain datasets focus on node-level or edge-level data, lacking in-depth graph-level or advanced hierarchical graph-level studies [13, 14]. Furthermore, most existing datasets are confined to single-chain data, which restricts the ability to compare and understand the complex characteristics of various blockchain systems [15, 16, 17, 18, 3].

Concurrently, the study of Graphs of Graphs (GoG), which captures intricate relationships and structures across various domains, is gaining traction. This framework is particularly beneficial in scenarios involving multiple levels of interaction or dependency, such as in chemical, social media,

---

[*]Corresponding author

38th Conference on Neural Information Processing Systems (NeurIPS 2024) Track on Datasets and Benchmarks.

and document collection domains [19, 20, 21]. Despite these advancements, most GoG datasets remain small and static, focusing predominantly on chemical and molecular interactions task [19, 22]. It is not clear whether existing GoG machine learning models can achieve satisfied performance in large-scale, real-life transaction networks.

To bridge these two gaps, we introduce novel datasets and a new GoG approach tailored to the blockchain ecosystem. On the blockchain, a wide variety of digital tokens represent diverse assets, such as DeFi tokens related to decentralized finance products and MEME tokens inspired by internet memes. While these tokens are distinct, they are interconnected, as they are implemented on the same blockchain and can interact with the same user groups. This interconnectivity allows us to design an advanced hierarchical approach that includes local graphs representing individual crypto token transactions and global graphs depicting inter-token relationships within the blockchain ecosystem. Our dataset covers two crucial aspects: first, it includes detailed label information for each token graph, categorizing tokens by behavior, including fraud identification; second, it integrates interactions between tokens across multiple blockchain platforms. Specifically, our dataset covers 268,282,924 transactions conducted by 18,600,142 cryptocurrency addresses, covering the transaction history of 24,316 tokens on three main EVM chains: Ethereum, Polygon, and Binance Smart Chain (BSC).

We conduct an in-depth analysis of the constructed GoG, employing systematic graph analysis and extensive machine learning techniques. Our findings indicate that tokens belonging to the same class can exhibit distinct graph characteristics, such as varying graph size, reciprocity, and clustering coefficient, depending on the blockchain they are on. Furthermore, tokens with a high number of edges in local graphs tend to possess high centrality in global graphs. Through experiments on machine learning models, we observe that methods based on GoG can outperform traditional GNN methods in anomaly detection, multi-class classification, and link prediction on blockchain graphs under specific conditions. However, we also note that existing GoG models often underperform in minor class classification, highlighting the need for more advanced techniques.

In summary, this work makes several key contributions:

- We introduce large-scale, cross-chain graphs-of-graphs datasets, enriching blockchain research with unprecedented depth of analysis.

- Our analysis of graph structures within the hierarchical approach reveals intriguing characteristics, underscoring the diversity of token graph structures across different chains.

- We investigate traditional graph machine learning models and GoG-based models in the datasets. Experimental results demonstrate that our datasets present new avenues and challenges for the blockchain and graph community.

## 2   Literature Review

**Blockchain Dataset.** A number of datasets have been developed for machine learning tasks on blockchain platforms. For instance, [15, 16, 17, 18] proposed Ethereum datasets specifically for account detection and link prediction. [23] introduced datasets for tokens and liquidity pools, conducting statistical analyses of tokenomics for Ethereum and BSC. [24] utilized both on-chain and off-chain data for predicting crypto trading prediction. In the realm of blockchain graph datasets, one pioneering benchmark is Chartalist [14], which encompasses multiple tasks across Bitcoin, Ethereum, and Dashcoin. However, the inherent differences in blockchain types, such as unspent transaction output (UTXO) and account-based systems, pose challenges for comparing tasks across these varied architectures. Recent studies have also focused on Ethereum's NFT markets. For example, [3] introduced a live graph lab for temporal graphs, facilitating the study of open, dynamic, and real transaction graphs from Ethereum NFT transactions. Moreover, [13] highlighted the significant role of linking on-chain Ethereum accounts with off-chain X accounts, emphasizing the value of off-chain data in enhancing Ethereum analysis. Despite these advancements, many studies remain focused on single chains, predominantly Ethereum, and concentrate on node-level or edge-level tasks. This narrow focus may limit the generalizability of their findings.

**Graph Representation Learning.** Graph representation learning transforms high-dimensional, sparse graph data into compact, dense vectors [25]. The main objective is to produce representation vectors that effectively capture both structural and feature information of extensive graphs [26]. Among various tasks in this field, one key task is graph classification, which focuses on predicting

the properties of whole graphs [27]. This task is widely used in social community analysis [28, 25] and molecular property prediction [29, 30]. Numerous GNN-based algorithms have been proposed to address graph classification [31, 32, 33, 34]. Generally, these algorithms employ the message-passing paradigm to iteratively refine node representations, followed by a graph pooling function to generate graph-level representations [35, 26].

**Graphs-of-Graphs.** Graphs-of-graphs (GoG) extends traditional graph theory by structuring individual graphs as nodes within a larger, interconnected graph. This structure enables the analysis of complex relationships between distinct graph-structured data [36]. Initial studies applied GoG to rank nodes in domain-specific networks [37] and developed clustering methods using non-negative matrix factorization (NMF) for multi-view and multi-domain graph clustering [21]. Later research applied the GoG approach to GNNs to enhance graph classification tasks [20, 38]. Recent efforts have furthered GoG models to improve prediction capabilities in chemical and drug interactions [19, 22, 39]. Additionally, [40] explored the use of multi-layer network models within Ethereum and Ripple for anomalous event analysis, demonstrating the effectiveness of similarly structured concepts in blockchain analytics. However, the application of GoG in blockchain datasets remains underexplored, indicating a potential area for further research.

In summary, we provide a detailed comparison of related and public datasets with our dataset in Table 1. Specifically, our dataset includes these unique features: (1) as a graphs-of-graphs dataset, it comprises large-scale local graphs, dense global graph structures, and real-life temporal edges; (2) as a blockchain graph dataset, it stands as the first large-scale hierarchical graphs-of-graphs dataset, encompassing multi-chain transactions and graph-level labels.

Table 1: Comparisons among open-source graphs-of-graphs and blockchain graph datasets with ours. The symbol "-" indicates data that is not related or applicable.

| Dataset | Field | Graph-level (token) labels | Multi-chain comparability | Density global graph | Avg. Num. local graph node | Avg. Num. local graph edge | Avg. global graph intra-inter edge | Dynamic |
|---|---|---|---|---|---|---|---|---|
| **Graphs-of-graphs datasets** | | | | | | | | |
| CCI900 [39] | Chemical | - | - | $4.4 \times 10^{-4}$ | 25.4 | 26.5 | 1707.3 | ✗ |
| CCI950 [39] | Chemical | - | - | $4.8 \times 10^{-4}$ | 26.2 | 27.4 | 511.4 | ✗ |
| NetBasedDDI [39] | Drug | - | - | $0.7 \times 10^{-1}$ | 24.8 | 26.7 | 442.2 | ✗ |
| ZhangDDI [39] | Drug | - | - | 0.3 | 25.2 | 27.0 | 1490.0 | ✗ |
| ChChMiner [39] | Drug | - | - | $0.5 \times 10^{-1}$ | 27.8 | 29.6 | 1418.9 | ✗ |
| DeepDDI [39] | Drug | - | - | 0.1 | 27.5 | 29.2 | 6153.8 | ✗ |
| Arxiv [38] | Text | - | - | $2.8 \times 10^{-4}$ | 30.9 | 200.1 | 23.31 | ✗ |
| QQ [38] | Social | - | - | $2.7 \times 10^{-3}$ | 291.2 | 2467.7 | 800.6 | ✗ |
| **Blockchain graph datasets** | | | | | | | | |
| Chartlist [14] | Blockchain | ✗ | ✗ | - | - | - | - | ✓ |
| LiveGraphLab [3] | Blockchain | ✗ | ✗ | - | - | - | - | ✓ |
| EX-Graph [13] | Blockchain | ✗ | ✗ | - | - | - | - | ✓ |
| Ours | Blockchain | ✓ | Ethereum | 0.3 | 1493.7 | 2225.2 | 14273.2 | ✓ |
| | | | Polygon | 0.5 | 1184.2 | 2523.6 | 525.1 | ✓ |
| | | | BSC | 0.6 | 1650.5 | 3346.4 | 4660.0 | ✓ |

## 3 Dataset Details

### 3.1 Background

**Blockchain and Cryptocurrency.** Blockchain technology operates on a decentralized network secured by cryptographically linked blocks. This structure ensures data immutability and verifiability, supporting secure, irreversible, and transparent transactions. Cryptocurrencies, built on this technology, facilitate secure digital transactions without a central authority, enhancing user anonymity while complicating fraud detection.

**EVM and ERC20.** The Ethereum Virtual Machine (EVM) supports an account-based model that enables direct value transfers and complex features like smart contracts. This model has become a standard for blockchain networks and decentralized applications, utilized across prominent networks such as Ethereum, Polygon, and Binance Smart Chain (BSC). The ERC20 standard on Ethereum and the BEP20 standard on BSC provide a framework for fungible tokens, promoting interoperability and simplifying the trading process across platforms.

**Accounts and Transactions.** On EVM-compatible chains, two primary account types exist: externally owned accounts (EOAs) and smart contracts. EOAs resemble traditional bank accounts but are controlled by individual private keys, while smart contracts are programmable accounts that execute automatically under specified conditions. Each account has a unique address, maintains a balance, and is controlled by a private key. Transactions include details such as the sender's and receiver's addresses, timestamps, values, and the transaction hash, ensuring that each transaction is immutable and traceable.

## 3.2 Data Collection

This section summarizes the statistics of our datasets, focusing on ERC20 tokens on Ethereum and Polygon, and BEP20 tokens on BSC. These token standards are among the most popular in the blockchain industry [41]. Our datasets include three independent sets, each targeting a different blockchain, comprising a total of 268,282,924 transactions conducted by 18,600,142 addresses across 24,316 tokens. These transactions record the full history of these tokens from the inception to February 2024. Detailed statistics are presented in Table 2.

Table 2: Statistics of the datasets.

| Chain | # Token | Start Month | End Month | # Transactions | # Addresses | # Categories |
|---|---|---|---|---|---|---|
| Ethereum | 14,464 | 2016-02 | 2024-02 | 81,788,211 | 10,247,767 | 290 |
| Polygon | 2,353 | 2020-08 | 2024-02 | 64,882,233 | 1,801,976 | 112 |
| BSC | 7,499 | 2020-09 | 2024-02 | 121,612,480 | 6,550,399 | 149 |

**Transaction Data.** All blockchain transactions are transparent, traceable, and publicly available, achieved through the secure linkage of blocks using cryptographic techniques [42]. Prominent blockchain explorers provide tools to easily access blockchain transaction data. We utilize public APIs from Etherscan[2], Polygonscan[3], and Bscscan[4] to facilitate the collection of token transactions. Each transaction includes sender and recipient addresses, transfer value, timestamp, unique transaction hash, and other relevant details.

**Tags.** We collect category tags of the tokens from the three prominent blockchain explorers as labels. We reviewed the tags for all ERC20 and BEP20 tokens launched no later than February 2024 across these platforms. We filtered the tokens to include only those with more than 10,000 addresses or 1,000,000 transactions to ensure fair data distribution. The label details are as follows:

- For fraud cases, we labeled tokens flagged by explorers as suspicious phishing or hack tokens. These include various kinds of spammed tokens, such as those that have been spammed to many users or those that pretend to be famous tokens, like fake USDT. In total, 7,198 fraud tokens were identified, representing 29.6% of the dataset.

- For other classes, we labeled tokens using the category tags given by the explorers. The most popular classes in the dataset include DeFi tokens, which are related to decentralized finance products; MEME tokens, often inspired by internet memes and characters; and Gaming tokens, which are associated with electronic gaming.

Figure 1a represents the diversity of categories within our dataset as a label cloud. In total, 313 categories are found in these tokens. However, the distribution of categories is very skewed, as shown in Figure 1b. Specifically, the top 5 categories on each chain can cover more than half of all tokens in our dataset.

## 3.3 Graph Construction

In this section, we present how we construct our Graphs of Graphs (GoG) datasets. We build two types of graphs: local graphs that represent transactions of tokens, and global graphs that represent token-token relationships. Figure 2 illustrates a sample of our GoG structure.

---

[2]`https://etherscan.io/`

[3]`https://polygonscan.com/`

[4]`https://bscscan.com/`

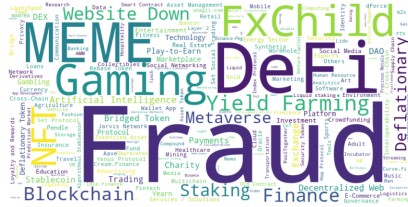

(a) Tokens categories label cloud.

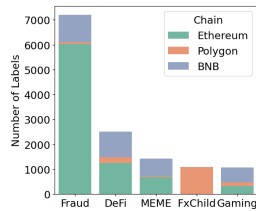

(b) Top 5 categories with highest total number.

Figure 1: Token categories analysis.

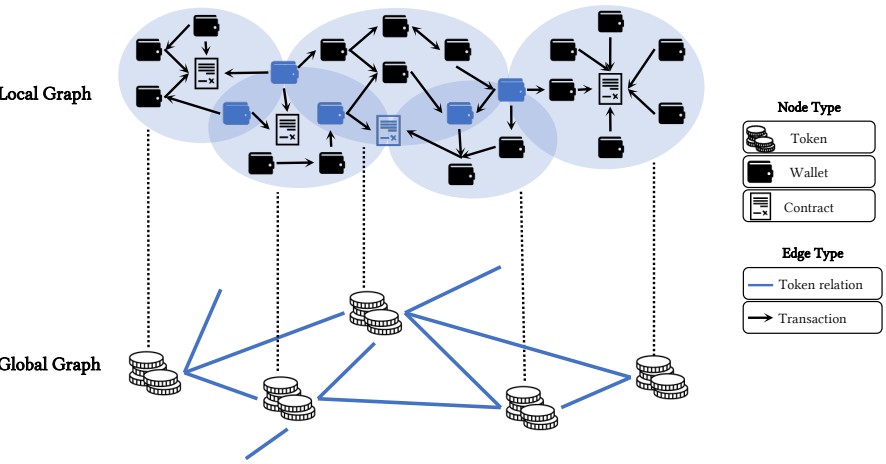

Figure 2: A sample GoG structure with 5 tokens. Blue wallets and contracts are involved in transactions of multiple tokens, while black wallets and contracts are involved in only one token's transactions.

### 3.3.1 Local Graph Construction

A local graph represents transactions of a single token. Defined as $G_{local} = (N_L, E_L)$, the graph consists of $|N_L|$ nodes representing accounts and $|E_L|$ edges representing transactions. Each edge ($e = (u, v, w, t)$) signifies a transaction from account $u$ to $v$, involving a value $w$ transferred at time $t$. The timestamp $t$ indicates when the transaction occurs, with the first transaction timestamp marking when the token becomes active on-chain.

### 3.3.2 Global Graph Construction

A global graph aims to model the correlation of various tokens across blockchain platforms. Specifically, we model the transaction networks of individual tokens into nodes, forming a graph $G_{global} = (N_G, E_G)$. $N_G$ denotes the set of all local graphs, and $E_G$ represents the inter-token relationships. The edges in the global graph are weighted by the Jaccard Coefficient, defined as:

$$\text{Edge weight: } J(A, B) = \frac{|A \cap B|}{|A \cup B|}$$

where $A$ and $B$ are the node sets of the local transaction graphs for two distinct tokens. The Jaccard Coefficient quantifies the degree of overlap in user bases between different tokens, offering insights into the inter-connectedness and user-sharing across tokens. Each edge weight reflects this inter-connectedness, providing a measure of relational strength and activity overlap between tokens at a global scale. This approach of finding common addresses is inspired by previous studies on social media groups [20] and multi-layer blockchain analysis [40]. We remove the null address[5] when measuring to prevent all tokens involved in transactions with null addresses from being connected. As new transactions are executed on the blockchain platforms, global edges in our graph dynamically adapt to changes in local transaction information, as detailed in Appendix D.

---

[5]Null address: 0x0000000000000000000000000000000000000000.

# 4 Observations and Analysis

## 4.1 Local Graph Analysis

We examine several graph properties within distinct classes of local graphs to deepen our understanding of token transfer networks. Figure 3 displays three crucial graph properties: **number of edges**, **reciprocity**, and **clustering coefficient**, segmented by the top five most prevalent classes across three blockchains. Notably, the "FxChild" class is exclusive to Polygon, while the other four classes are observed across all chains.

> **Finding (1)**: The distribution of token categories varies across different chains. Tokens within the same class can exhibit distinct network characteristics depending on the blockchain.

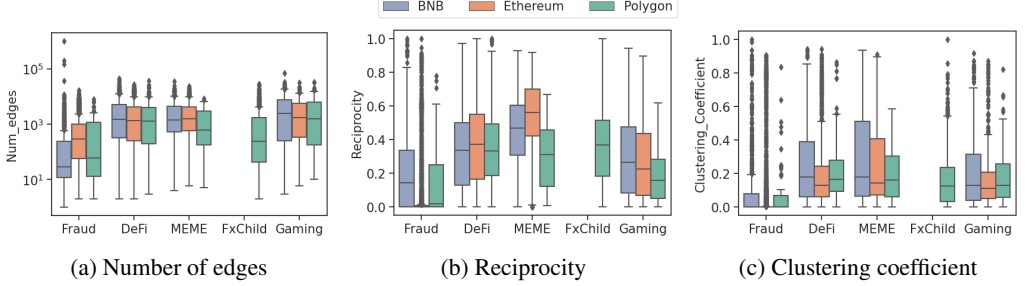

| (a) Number of edges | (b) Reciprocity | (c) Clustering coefficient |

Figure 3: Distribution of graph properties across three chains.

Figure 3a illustrates the distribution of the number of edges across token graphs on three blockchains. While DeFi, MEME, and Gaming tokens generally show a comparable number of edges, Fraud tokens consistently exhibit fewer edges across all platforms, with a notably higher node count on Polygon. This suggests less connectivity among participants in fraud-related activities.

Figure 3b shows the reciprocity of these token graphs, reflecting the proportion of mutual connections. Results indicate that MEME tokens display higher reciprocity, aligning with the interactive nature of these communities. In contrast, Fraud tokens show the lowest reciprocity, indicating that fraudulent transactions are less likely to be reciprocal, possibly due to their unilateral nature.

Figure 3c presents the clustering coefficient, which indicates how closely nodes in a graph cluster, reflecting the formation of tight-knit groups or collusive clusters. A higher average clustering coefficient suggests the presence of prevalent cliques or active trading communities. Fraud tokens demonstrate the lowest clustering coefficients, suggesting sparse connectivity, whereas MEME tokens exhibit the highest, indicative of tight-knit communities. Additionally, tokens on BSC display the widest range and highest clustering coefficients compared to those on Ethereum and Polygon, pointing to more clustered network structures on BSC.

## 4.2 Global Graph Analysis

To understand how the Graph of Graphs (GoG) approach enhances our comprehension of the intricate relationships and interactions within the ERC20 markets, we perform sophisticated network analyses, focusing on **edge weight analysis** and **node centrality** to identify influential tokens.

> **Finding (2):** The predominance of low edge weights across the network suggests limited interaction between different tokens. High weights are predominantly observed among local graphs within the same class, especially those implicated in fraudulent activities.

Edge weights in the global graph, determined by the Jaccard coefficient of common nodes, illustrate the interconnectedness of tokens based on shared investors or smart contracts. We analyze the distribution of edge weights in the global networks, as demonstrated in Figure 4. This distribution is predominantly characterized by small values, indicating sparse connections across most token pairs, although there are exceptions with a few highly interconnected node pairs. Moreover, as shown in Table 3, the contract pairs with the highest weights on each blockchain consistently involve local graphs within the same class, notably marked by a prevalence of fraud-related contracts. This pattern suggests concentrated activity or potential collusive behaviors within these groups of tokens.

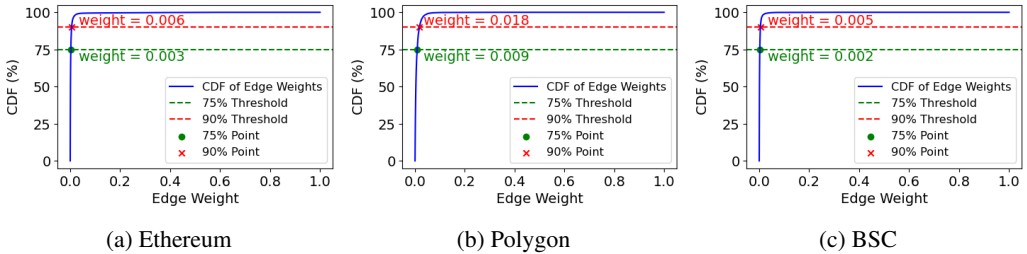

(a) Ethereum  (b) Polygon  (c) BSC

Figure 4: Cumulative distribution of edge weights in three global graphs.

Table 3: Top 3 connected contracts with highest weight across different chains.

| Chain | Contract1 | Contract2 | Class1 | Class2 | Edge Weight |
|---|---|---|---|---|---|
| Ethereum | 0x3d09...c61cce | 0x6752...1e761a | Fraud | Fraud | 1.0 |
| | 0xa034...145c1b | 0x5fbf...762f24 | Fraud | Fraud | 1.0 |
| | 0xb3f6...7d8429 | 0x6249...29af88 | Deprecated | Deprecated | 1.0 |
| Polygon | 0x8db0...06f7ec | 0x36f5...c72c5b | FxChild | FxChild | 1.00 |
| | 0xbbcc...85e429 | 0x1a8a...6f5f68 | FxChild | FxChild | 0.98 |
| | 0xa7e8...9304f7 | 0xee35...52699c | Gaming | Gaming | 0.97 |
| BSC | 0xaf71...020ac2 | 0x362d...881ffc | Play-to-Earn | Play-to-Earn | 1.0 |
| | 0xaefe...55eab2 | 0x9775...4e74ec | Fraud | Fraud | 1.0 |
| | 0x5fb0...42a2e4 | 0x9775...4e74ec | Fraud | Fraud | 1.0 |

**Finding (3):** A higher number of edges in local graphs typically correlates with central roles in global graphs, highlighting a strong relationship between transaction activity and centrality within the blockchain ecosystem.

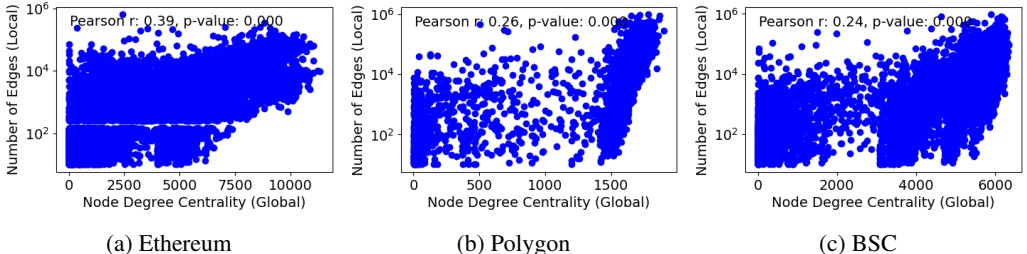

(a) Ethereum  (b) Polygon  (c) BSC

Figure 5: Global node degree centrality vs. local graph number of edge.

Node degree centrality is a critical metric for identifying hub nodes—those that are most interconnected within the market. We analyze node degree centrality within each global graph, aiming to identify tokens that act as central nodes in their respective markets. First, we find that tokens with high node degrees tend to belong to the most popular classes in the blockchain. More than 70% of the top 10 nodes with the largest degree centrality belong to the most popular classes, including Fraud, FxChild, and Gaming. Second, we explore the correlation between node degree centrality in global graphs and the number of edges in local graphs, as illustrated in Figure 5. Results reveal that token graphs with a higher number of edges are more likely to become high-degree nodes in global graphs. Interestingly, we found that very few tokens have between 150 and 250 transactions, which creates a noticeable gap in the Figure 5a.

## 5   Downstream Applications

The observations presented earlier provide a comprehensive overview of our GoG datasets. Our findings highlight how the GoG approach enhances our understanding of the complex transfer graphs across various token classes. These properties introduce new challenges for several downstream tasks. In this section, we investigate two studied tasks to address these questions:

- Q1: How can the GoG approach improve accurate classification of token graph categories?

- Q2: How can the GoG approach improve fraud detection performance on blockchain graphs?

- Q3: How do graph learning models perform across different blockchain datasets?

Additionally, we explore investment prediction as a future edge prediction task in Appendix F. These experiments demonstrate the capability of the GoG approach to handle various graph learning tasks in the blockchain domain, thereby underscoring its potential for practical applications.

## 5.1 Multi-class Graph Classification

Graph classification is a crucial aspect of graph learning research [43]. By predicting the attributes of each graph within a collection, graph classification facilitates exploration across various domains, including image classification [44], document analysis [45], and chemical discovery [46]. In this section, we focus on multi-class graph classification, aiming to categorize tokens into distinct classes. Given the scarcity of some minor classes, we concentrate on the top token categories within each chain for our classification task. Specifically, we classify tokens into two groups: (1) the top 3 categories, and (2) the top 5 categories.

**Models.** We compare two groups of models. Group 1 consists of GNN models applied to individual graphs, including: (1) GCN [31], (2) GAT [47], (3) GIN [48], (4) ResidualGCN [49], and (5) GraphSage [50]. Group 2 comprises GNN models tailored for handling collections of graphs, including: (1) SEAL [20], which applies a self-attentive graph embedding approach using GCN as the base model to embed individual graph instances into fixed-length vectors for classification; (2) GoGNN [19], which enhances GCN's capabilities by incorporating an attention-based pooling mechanism and GAT to effectively identify key substructures within local graphs; and (3) DVGGA [39], which combines a denoising autoencoder with a self-attentive GNN and readout function.

**Settings.** We filter out token graphs with fewer than five nodes or edges to maintain data integrity. After applying this criterion, less than 2% of tokens were removed from all three datasets, ensuring that our analysis still represents the majority of the data. Guided by insights from subsection 4.2, we establish a threshold for edge weight, including only weights exceeding 0.01 to identify closely connected tokens. When experimenting with both groups of models, we utilize incoming degree, outgoing degree, and total degree as node features for the local graphs. For a fair comparison, we conduct all experiments as supervised learning tasks. The dataset is divided into training and testing sets following an 80/20 ratio. Then, we employ Macro-F1 and Micro-F1 as evaluation metrics. Macro-F1 computes the F1-score separately for each class and then averages them, giving equal importance to all classes regardless of their frequency. In contrast, Micro-F1 assigns more weight to classes with higher frequencies, reflecting their prevalence in the dataset.Due to the imbalance in our dataset, we primarily use Macro-F1 for model comparisons. Each experiment is repeated three times with different seeds, and we report the average performance and standard deviation.

**Results.** The results of the 3-class and 5-class classification tasks are summarized in 3. Several notable observations emerge from these results. First, GoG models exhibit superior performance compared to individual GNN models across most tasks in both classification scenarios. Specifically, SEAL demonstrates the best F1-macro performance, showing up to 28% and 16% improvements respectively in the 3-class category on BSC, and up to 44% and 11% improvements respectively in the 5-class category on Ethereum, compared to the average performance of non-GoG models. Second, as the classification task becomes more complex by including additional minor classes, the performance of both model groups notably declines. The advantage of GoG models over individual GNN models diminishes in 5-class classification compared to 3-class classification, emphasizing the necessity for further development of advanced GoG models, especially for minor-class classification. Third, all model groups demonstrate less satisfactory performance on the Polygon dataset. This could be attributed to the dataset's smaller size and greater imbalance compared to others. Therefore, there is a clear need for devising robust graph learning models capable of effectively capturing the intricacies of the Polygon dataset. In Appendix G, we conducted experiments predicting the class label of younger tokens using the information about older tokens. Results show that for Ethereum and BNB, the performance shows slight differences from the results in Table 4. However, for Polygon, the performance deteriorates significantly.

Table 4: 3-class and 5-class classification performance by blockchain.

| Model | Ethereum | | Polygon | | BSC | |
|---|---|---|---|---|---|---|
| | F1-macro | F1-micro | F1-macro | F1-micro | F1-macro | F1-micro |
| 3-Class Classification | | | | | | |
| GCN | $62.48_{\pm6.31}$ | $85.05_{\pm1.38}$ | $28.82_{\pm1.86}$ | $74.24_{\pm0.83}$ | $51.43_{\pm5.93}$ | $57.02_{\pm3.37}$ |
| GAT | $60.22_{\pm7.04}$ | $84.62_{\pm1.23}$ | $29.90_{\pm2.60}$ | $73.94_{\pm1.79}$ | $54.48_{\pm6.15}$ | $59.96_{\pm3.19}$ |
| GIN | $39.79_{\pm11.02}$ | $78.58_{\pm3.07}$ | $28.82_{\pm1.53}$ | $74.26_{\pm0.83}$ | $43.29_{\pm2.93}$ | $55.90_{\pm2.86}$ |
| ResidualGCN | $62.85_{\pm6.07}$ | $84.18_{\pm1.50}$ | $28.50_{\pm0.35}$ | $74.37_{\pm0.18}$ | $50.73_{\pm4.59}$ | $56.78_{\pm2.29}$ |
| GraphSage | $64.17_{\pm8.53}$ | $85.51_{\pm2.05}$ | $31.71_{\pm2.56}$ | $74.48_{\pm0.68}$ | $56.70_{\pm6.12}$ | $61.36_{\pm2.78}$ |
| SEAL | $\mathbf{67.31}_{\pm3.60}$ | $\mathbf{86.65}_{\pm1.30}$ | $29.64_{\pm1.70}$ | $\mathbf{74.51}_{\pm0.16}$ | $\mathbf{63.77}_{\pm0.59}$ | $\mathbf{65.59}_{\pm0.42}$ |
| GoGNN | $64.20_{\pm4.29}$ | $85.89_{\pm0.47}$ | $\mathbf{36.11}_{\pm0.50}$ | $66.09_{\pm11.02}$ | $53.98_{\pm4.55}$ | $58.03_{\pm2.90}$ |
| DVGGA | $37.23_{\pm10.57}$ | $77.84_{\pm4.16}$ | $28.44_{\pm0.004}$ | $74.22_{\pm0.17}$ | $41.31_{\pm8.67}$ | $47.03_{\pm7.64}$ |
| 5-Class Classification | | | | | | |
| GCN | $36.88_{\pm4.90}$ | $78.37_{\pm1.81}$ | $16.40_{\pm0.65}$ | $68.79_{\pm0.66}$ | $27.49_{\pm3.27}$ | $43.09_{\pm2.12}$ |
| GAT | $36.46_{\pm4.44}$ | $78.46_{\pm1.42}$ | $\mathbf{20.04}_{\pm3.54}$ | $68.85_{\pm1.08}$ | $29.85_{\pm3.12}$ | $44.97_{\pm2.77}$ |
| GIN | $19.24_{\pm4.62}$ | $71.31_{\pm2.18}$ | $16.41_{\pm0.65}$ | $68.82_{\pm0.63}$ | $22.01_{\pm2.32}$ | $41.33_{\pm3.19}$ |
| ResidualGCN | $33.72_{\pm5.56}$ | $76.69_{\pm1.91}$ | $16.47_{\pm0.86}$ | $68.6_{\pm12.30}$ | $23.82_{\pm5.09}$ | $40.33_{\pm3.90}$ |
| GraphSage | $38.91_{\pm5.31}$ | $79.73_{\pm1.97}$ | $18.01_{\pm1.64}$ | $68.88_{\pm0.65}$ | $30.22_{\pm3.34}$ | $46.15_{\pm2.20}$ |
| SEAL | $\mathbf{45.09}_{\pm10.79}$ | $\mathbf{81.59}_{\pm2.06}$ | $16.90_{\pm0.82}$ | $\mathbf{69.02}_{\pm0.15}$ | $\mathbf{31.83}_{\pm3.31}$ | $\mathbf{46.50}_{\pm1.93}$ |
| GoGNN | $32.51_{\pm3.58}$ | $71.74_{\pm6.56}$ | $19.70_{\pm2.62}$ | $59.19_{\pm12.41}$ | $23.33_{\pm8.42}$ | $38.54_{\pm6.30}$ |
| DVGGA | $21.60_{\pm7.21}$ | $71.70_{\pm2.29}$ | $18.03_{\pm2.38}$ | $67.20_{\pm2.42}$ | $16.92_{\pm5.77}$ | $34.91_{\pm4.89}$ |

## 5.2 Graph Anomaly Detection

Anomaly detection is a significant task in machine learning with numerous applications, including anti-money laundering [51], social media analysis [52], and disease detection [53]. In this section, we focus on detecting anomalies in tokens, specifically identifying fraudulent tokens from non-fraudulent ones. Given the skewness of fraud and non-fraud tokens, we approach graph anomaly detection as an unsupervised learning task.

**Models.** We compare two groups of models. Group 1 includes anomaly detection methods for multivariate data, such as probabilistic and outlier ensemble methods. Specifically, we compare (1) COPOD [54], (2) IForest [55], (3) DIF [56], and (4) VAE [57]. Group 2 includes anomaly detection methods on graphs, primarily GNN+AE methods. Specifically, we compare (1) GAE [58], (2) DONE [59], (3) DOMINANT [60], (4) AnomalyDAE [61], and (5) CoLA [62]. Detailed introductions of these methods are presented in Appendix E.

**Settings.** We use the same dataset settings as in subsection 5.1 to filter out small token graphs and build global graphs. For multivariate data analysis, drawn from our observations in subsection 4.1, we measure various graph properties, including the number of nodes, edges, assortativity, density, and reciprocity. These features are normalized to ensure consistency across the dataset. Our experimental setup follows the frameworks provided by PyOD [63] and PyGOD [64]. The dataset is divided into train/validation/test sets following an 80/10/10 ratio. Evaluation metrics include the area under the curve (AUC) and Average Precision (AP). AUC measures the model's ability to rank anomalies higher than normal instances, while AP quantifies the precision-recall balance, providing insights into model performance regarding the anomaly detection task.

**Results**. The performance of graph anomaly detection methods across three blockchains is summarized in Table 7. Interestingly, on the BSC dataset, most graph outlier detection methods outperform those based on graph structural data. For example, AnomalyDAE shows up to 3.34% improvement in AUC and 54.20% improvement in AP, compared to the average performance of detection models based on graph structural data. However, on Ethereum and Polygon, methods based on graph structural data demonstrate superior performance. This variation may be attributed to differences in fraudulent token behaviors and network structures specific to each blockchain. Additionally, consistent with the findings in subsection 5.1, both groups of methods exhibit poorer performance on the Polygon dataset, highlighting the need for further research. In Appendix H, we explored an additional method to represent token graphs by employing the DeepWalk algorithms [65]. Results show that

Table 5: Graph anomaly detection performance by blockchain. We report the ratio of number of non-fraud:fraud case of each data at the top.

| Model | Ethereum (8387: 6022) | | Polygon (2257: 58) | | BSC (6339: 1042) | |
|---|---|---|---|---|---|---|
| | AUC | AP | AUC | AP | AUC | AP |
| COPOD | $83.27_{\pm1.09}$ | $27.25_{\pm0.4}$ | $60.52_{\pm13.27}$ | $\mathbf{11.33} \pm 6.49$ | $52.87_{\pm2.09}$ | $14.18_{\pm0.69}$ |
| IForest | $84.10_{\pm0.55}$ | $26.93_{\pm0.56}$ | $64.33_{\pm11.43}$ | $10.79_{\pm5.67}$ | $58.36_{\pm2.83}$ | $11.58_{\pm1.57}$ |
| DIF | $\mathbf{84.56} \pm 1.31$ | $32.69_{\pm0.95}$ | $68.04_{\pm10.11}$ | $7.99_{\pm2.06}$ | $51.57_{\pm0.49}$ | $17.52_{\pm2.05}$ |
| VAE | $67.25_{\pm1.61}$ | $31.46_{\pm0.49}$ | $\mathbf{72.45} \pm 10.41$ | $10.56_{\pm5.09}$ | $59.03_{\pm0.20}$ | $18.70_{\pm1.13}$ |
| GAE | $70.85_{\pm2.58}$ | $31.21_{\pm0.68}$ | $62.16_{\pm0.09}$ | $3.85_{\pm0.01}$ | $56.33_{\pm1.25}$ | $17.11_{\pm0.35}$ |
| DONE | $74.93_{\pm2.91}$ | $29.03_{\pm0.92}$ | $62.21_{\pm0.30}$ | $1.95_{\pm0.07}$ | $65.86_{\pm3.70}$ | $10.64_{\pm1.10}$ |
| DOMINANT | $75.18_{\pm2.69}$ | $\mathbf{43.14} \pm 19.69$ | $70.45_{\pm7.93}$ | $3.55_{\pm1.48}$ | $\mathbf{78.87} \pm 0.23$ | $8.49_{\pm0.03}$ |
| AnomalyDAE | $65.82_{\pm8.47}$ | $39.24_{\pm10.09}$ | $60.94_{\pm3.06}$ | $3.72_{\pm0.42}$ | $62.49_{\pm9.23}$ | $\mathbf{22.71} \pm 6.98$ |
| CoLA | $65.15_{\pm7.17}$ | $35.80_{\pm7.04}$ | $54.90_{\pm2.74}$ | $3.51_{\pm0.64}$ | $60.87_{\pm3.63}$ | $19.64_{\pm6.29}$ |

while GoG models benefit from the use of the DeepWalk algorithm, the performance of multivariate outlier detection methods decreases. In general, the adoption of the GoG approach presents new opportunities to enhance graph anomaly detection in the blockchain domain, as evidenced by the varied performance observed across different blockchains.

# 6 Conclusion

In this paper, we introduced a novel dataset based on the Graphs of Graphs (GoG) approach within the blockchain domain. Our dataset includes local graphs that detail individual token transactions and global graphs that model interactions between tokens across multiple blockchain platforms. This approach provides a comprehensive view of transaction activities within the blockchain ecosystem. We conducted systematic analyses and experiments using the GoG approach, revealing significant patterns and characteristics in the blockchain environment. Our findings suggest that GoG models have the potential improve various applications, such as link prediction, anomaly detection, and token classification, especially when compared to traditional GNN methods. We believe this work lays a foundation for future research in graph learning and encourages further exploration of the complex relationships within blockchain networks.

# 7 Border Impact and Limitation

Our datasets offer researchers fresh opportunities to explore and analyze blockchain graphs. Insights from this analysis could aid in developing stronger market structures and enhanced security protocols within crypto token platforms. Our classification and anomaly detection models aim to enhance predictive capabilities, particularly in situations where labels may be incomplete or unavailable. For instance, while only a small fraction of existing tokens has been labeled by blockchain explorers, there are over 900,000 ERC20 tokens on Ethereum, with new tokens being launched daily that may not yet be classified. Our models can predict these labels more quickly and provide detailed insights. By leveraging similarities within the data, the GoG approach reduces reliance on labeled data and improves the accuracy of predictions for unlabeled tokens.

However, our datasets have several limitations that must be acknowledged. First, the lack of restrictions on creating multiple accounts on the same blockchain allows a single entity to control multiple accounts. This can distort interaction patterns and connectivity metrics within our GoG datasets. Second, while our model is designed to enhance predictive capabilities in cases where labels are incomplete, it is limited by the fact that only a small fraction of existing tokens has been labeled by blockchain explorers. Although we rely on trusted blockchain explorers for labeling, there is always a risk of misclassification, especially with fraudulent tokens. Therefore, our model is intended to augment, not replace, human judgment; predictions should be viewed as suggestions that require further scrutiny. Third, the public nature of blockchain data raises privacy concerns. Our datasets link transactions to wallet addresses, potentially enabling the tracking of individual behaviors, which could lead to targeted advertising or surveillance.

## Acknowledgments and Disclosure of Funding

This research is supported by the National Research Foundation, Singapore under its Industry Alignment Fund – Pre-positioning (IAF-PP) Funding Initiative. Any opinions, findings and conclusions or recommendations expressed in this material are those of the author(s) and do not reflect the views of National Research Foundation, Singapore. The authors would like to thank reviewers for their helpful comments.

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

# A   Background

In this section, we provide an overview of blockchain technology and cryptocurrency, laying the groundwork for understanding the subsequent discussions in this paper.

**Blockchain and Cryptocurrency.** Blockchain technology has gained growing attention recently for its strong security features and decentralized structure. It is characterized by a sequence of cryptographically secured blocks that operate on a network of nodes [42]. This design ensures data immutability and verifiability while allowing universal access, enabling participants to interact with the ledger from anywhere at any time. Once recorded on the ledger, transactions become irreversible and are executed securely and transparently, which helps safeguard the integrity of data exchanges.

With the support of blockchain technology, cryptocurrencies have surged in popularity as an innovative means of conducting secure digital transactions. Unlike traditional currencies, cryptocurrencies operate without a centralized authority and are managed through decentralized systems. This decentralization maintains participant anonymity, offering robust privacy protection; however, it complicates efforts to identify fraudulent activities within the market.

**Blockchain Models and EVM Chains.** Various operational models exist within blockchain technology. For instance, Bitcoin, the first cryptocurrency network, operates using the Unspent Transaction Output (UTXO) model [66]. In this model, each transaction utilizes unspent outputs from previous transactions as inputs, generating new unspent outputs for subsequent transactions. This method preserves transaction integrity by streamlining ownership verification and enhancing security measures related to transaction immutability.

In contrast, the Ethereum Virtual Machine (EVM) introduced an account-based model, akin to traditional banking systems, where balances are maintained in user accounts [67]. This model enables direct value transfer and supports advanced features such as smart contracts, which are self-executing agreements with terms embedded directly within the blockchain. Due to its versatility and strong developer support, the EVM has become the standard for building blockchain networks and decentralized applications. The three notable EVM-based networks discussed in this work are Ethereum, Polygon, and Binance Smart Chain [68]:

- *Ethereum*, the pioneering EVM chain, has developed a robust platform for decentralized applications. It supports a wide range of decentralized services, from financial transactions to games and autonomous organizations. Its native token, Ether, holds the second-largest market capitalization, second only to Bitcoin.

- *Polygon* enhances Ethereum's functionality as an EVM-compatible chain by offering faster transactions and reduced fees. Functioning as a sidechain to Ethereum, it addresses scalability issues with a multi-chain infrastructure, which is particularly advantageous for developers seeking efficient transaction throughput within the Ethereum ecosystem.

- *Binance Smart Chain* provides a similar EVM-compatible environment with a focus on scalability and user experience. It has carved out a niche by emphasizing rapid transactions and minimal fees, particularly attracting decentralized finance (DeFi) applications and NFTs.

**ERC20 and BEP20 Standards.** The ERC20 standard defines a framework for fungible tokens on the Ethereum blockchain. These fungible tokens are digital assets that are identical in type and value, making them interchangeable with one another. This standardization simplifies the process of trading and exchanging tokens and enhances their interoperability across various applications. Similarly, BEP20 is a standard used on the Binance Smart Chain (BSC), mirroring many of the functionalities of ERC20 while optimizing for faster transactions and lower fees.

**Accounts and Transactions.** EVM-compatible chains typically support two principal types of accounts: External Owned Accounts (EOAs) and smart contracts. EOAs function much like traditional bank accounts, as they are directly managed by users through a private key, granting them full autonomy over transactions. In contrast, smart contracts are autonomous programs that reside on the blockchain and execute automatically when predefined conditions are met. These programs are crucial for a variety of operations on EVM chains, from facilitating transactions in the token markets to managing decentralized finance (DeFi) protocols and automated governance mechanisms.

A transaction includes various details, such as the sender's and recipient's actions, signature, nonce, data, gas limit, maximum priority fee per gas, and maximum fee per gas. In the token market, these

transactions facilitate diverse blockchain events like token issuance and transfers. This architectural framework not only supports complex financial interactions but also enhances security across the blockchain ecosystem.

## B    Supplemental Related Work

**Graphs-of-Graphs Analysis.** The analysis of Graphs-of-Graphs (GoG) systems has become a crucial method for understanding complex relationships within and across different network layers in various domains. For instance, Chen et al. [69] examined the dynamics of event propagation on social platforms like Twitter. They analyzed follower link roles by grouping users based on their language settings, treating these groups as local graphs, with following or retweeting relationships represented as edges. Similarly, Wang et al. [70] modeled intra-level and inter-level causal relationships within interdependent networks, effectively tracing and identifying root causes in complex interconnected system structures. In more specialized applications, Liu et al. [71] employed GoG to enhance hazard identification at construction sites. They mapped interactions between characters and hazard networks, simplifying complex network structures to improve safety outcomes. Additionally, Chen et al. [72, 73] investigated the manipulation of connectivity in multi-layered networks, uncovering the structural dynamics that govern these complex systems. These studies underscore the powerful capability of GoG analysis in providing a deeper understanding of intricate graph systems.

## C    Basic Structure Properties

In this section, we explore several fundamental graph properties relevant to our analysis, as discussed in subsection 4.1 and subsection 5.2. We measure seven key graph properties: the number of nodes, the number of edges, density, assortativity, reciprocity, clustering coefficient, and effective diameter. These properties provide a comprehensive structural overview of the graph, which is essential for understanding its characteristics and implications in the context of token transfer networks.

First, we consider the number of nodes and edges, which quantitatively describe the scale and potential complexity of the graph. Density, assortativity, and reciprocity offer insights into the connectivity and interaction patterns among nodes, reflecting how edges are distributed and whether similar nodes preferentially connect to each other. Additionally, the clustering coefficient and effective diameter provide a view of the overall compactness and reachability within the graph.

**Density.** The density of a graph measures its compactness and connectivity. In this study, density is calculated as:
$$D = \frac{|E|}{|V|(|V| - 1)}$$
where $|E|$ is the number of edges, and $|V|$ is the number of nodes. In token transfer graphs, a lower density suggests a fragmented or developing market, indicative of fewer interactions or participants. Conversely, a high density indicates a mature market with frequent transactions between participants. This distinction is crucial for understanding market dynamics.

**Assortativity.** The assortativity coefficient quantifies the tendency of nodes to connect with others that share similar attributes. Specifically, assortativity is calculated by:
$$r = \frac{\sum_{(i,j) \in E} (f(i) - f_1)(f(j) - f_2)}{\sqrt{\sum_{(i,j) \in E} (f(i) - f_1)^2 \sum_{(i,j) \in E} (f(j) - f_2)^2}}$$

This metric is particularly relevant in token transfer graphs, as it measures how frequently addresses transact with others of similar characteristics. A higher assortativity may indicate a market dominated by similar types of transactions or participants. However, it is important to note that this is a trend observed in our data rather than an absolute rule. Understanding this property aids in identifying market segmentation.

**Reciprocity.** Reciprocity measures the likelihood of directed connections being reciprocated. It is calculated by:
$$\rho = \frac{|\{(i,j) \in G : (j,i) \in G\}|}{|E(G)|}$$

This metric is crucial for understanding mutual interactions between addresses, such as reciprocal trading patterns. In token transfer graphs, a higher reciprocity suggests a strong bidirectional transactional relationship, indicating trust or partnership between nodes. This insight is vital for assessing the stability of relationships within the graph.

**Clustering Coefficient.** The clustering coefficient measures how closely nodes in a graph tend to cluster together. This metric is essential in token transfer graphs, as it indicates the extent to which nodes form tightly-knit groups, which may suggest collusive behavior or strong community structures. We primarily use the average clustering coefficient to assess overall network cohesion and the potential for collaborative behavior among participants. It is calculated as:

$$C_{\text{avg}} = \frac{1}{n} \sum_{i=1}^{n} C_i$$

$$C_i = \frac{2T(i)}{k_i(k_i - 1)}$$

In the token transfer graph, a higher average clustering coefficient suggests a network characterized by prevalent cliques or groups that engage in frequent interactions, potentially indicating tight-knit trading communities.

**Effective Diameter.** The effective diameter provides insight into the average separation between node pairs across the graph. We measure the effective diameter by performing breadth-first search (BFS) from a sample of randomly selected nodes to provide a broad and representative overview of the graph's structure. The effective diameter is then defined as the 90th percentile of the shortest path lengths obtained from these BFS runs. This approach estimates how far apart nodes are on average, considering the most representative paths rather than extremes. The effective diameter reflects how easily a token can circulate within the network, a key factor in assessing liquidity and market efficiency. This metric is particularly important for understanding the graph's accessibility.

## D    Temporal Properties Analysis

To reveal the temporal changes in the GoG systems of the three blockchains, we analyze the yearly variation of some fundamental properties of the global graphs. Nodes represent tokens, and an edge between two nodes indicates that the tokens share common addresses during that year.

First, we examine the dynamics of the number of nodes and edges, as illustrated in Figure 6. Across the Ethereum, Polygon, and BSC ERC20 token networks, we observe a consistent trend of significant growth in both nodes and edges. This growth reflects increased adoption and diversification of blockchain platforms. Over the past three years, the average increase in the number of nodes in the global graphs is 42.49% for Ethereum, 33.08% for Polygon, and 65.18% for BSC. These figures indicate substantial changes in the dataset. Notably, Ethereum exhibits the most mature growth pattern, particularly with a significant acceleration since 2020. In contrast, Polygon shows robust growth; however, it has a slower increase in edges compared to nodes, suggesting a less interconnected network than Ethereum's GoG. Meanwhile, BSC experiences a rapid rise in both nodes and edges but begins to show signs of stabilization in 2023, indicating a maturing of its initial expansion phase. These patterns highlight that while all networks are expanding, the nature and rate of growth vary among the different blockchains.

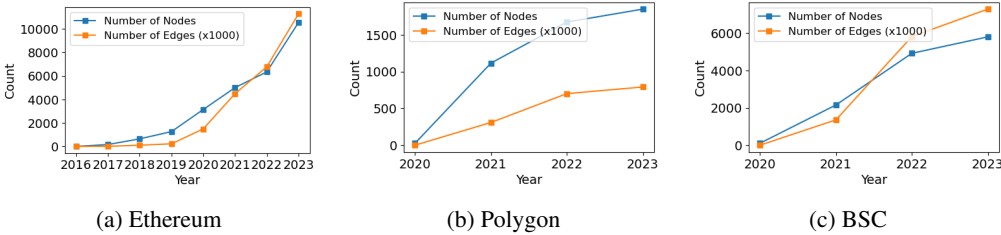

|          (a) Ethereum          |          (b) Polygon          |          (c) BSC          |

Figure 6: Yearly number of nodes and edges of three global graphs.

Second, we analyze the density and average clustering coefficient of the three global graphs, as shown in Figure 7. A common trend emerges across Ethereum and BSC: both density and clustering

coefficient tend to decrease as the network size increases. This trend indicates sparser connections as these networks expand, especially pronounced in the BSC network, which reflects significant diffusion from its originally dense structure. Conversely, Polygon exhibits a different pattern; both metrics initially increase and then stabilize. This indicates that the GoG not only grows but also effectively maintains or enhances its clustering. Such behavior suggests robust internal structuring that preserves community integrity even as the network scales. These observations highlight varied adaptive strategies within blockchain networks, with the Polygon GoG notably sustaining community cohesion amidst growth.

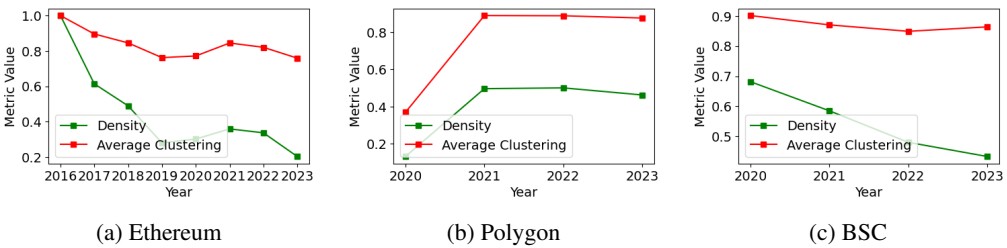

(a) Ethereum        (b) Polygon        (c) BSC

Figure 7: Yearly density and clustering coefficients of three global graphs.

# E  Model Implementation Details

In this section, we introduce the models and hyperparameters we used for the multi-class classification and anomaly detection tasks.

## E.1  Multi-class classification

**Models.** We conduct experiments on two groups of models: (1) 5 GNN models on individual graphs, and (2) 3 GoG-based GNN models on graphs-of-graphs.

Group (1) includes GNN models for individual graphs:

- *Graph Convolutional Network (GCN)* [31] utilizes a layer-wise propagation rule based on spectral graph convolutions, enabling it to learn representations that capture graph structure and node features effectively.
- *Graph Attention Network (GAT)* [47] introduces an attention mechanism in the propagation step, allowing nodes to dynamically weigh the contributions of their neighbors.
- *Graph Isomorphism Network (GIN)* [48] is designed to capture the power of the Weisfeiler-Lehman graph isomorphism test. It approximates the Weisfeiler-Lehman graph isomorphism test by adjusting aggregators to better distinguish between different graph structures.
- *Residual Graph Convolutional Network (ResidualGCN)* [49] incorporates residual connections into the graph convolutional layers to improve gradient flow during training, which enhances the learning of deeper GNN architectures by mitigating the vanishing gradient problem.
- *GraphSAGE* [50] generates node embeddings by sampling and aggregating features from a node's local neighborhood. Its inductive learning framework supports embedding generation for unseen data, making it scalable and efficient for large graphs.

Group (2) includes models designed for graphs-of-graphs:

- *Semi-supervised Graph Classification via Cautious Iteration (SEAL)* [20] utilizes a self-attentive graph embedding method with GCN as a backbone to embed graph instances into fixed-length vectors, facilitating graph-based classification tasks. It enhances the encoding of local graph structures and their relationships within a larger graph context.
- *Graph of Graphs Neural Network (GoGNN)* [19] extends traditional GCN capabilities by integrating an attention-based pooling mechanism and GAT. It effectively identifies significant substructures within local graphs and interactions within the interaction graph, providing a powerful framework for analyzing complex graph relationships.

- *Denoising Variational Graph Autoencoder (DVGGA)* [39] employs a denoising variational autoencoder combined with a self-attentive graph neural network and a readout operation. This model is adept at handling noise in graph data, making it suitable for tasks requiring robust feature extraction and anomaly detection in noisy environments.

**Model structures.** For GNN models targeting individual graphs, we employ a configuration that includes two GNN layers followed by a fully connected layer for classification. This two-layer setup, consistent with the backbone design of SEAL [20], ensures fair comparisons. Each layer transforms node features to enhance feature extraction, using ReLU activation and dropout for regularization. Following the convolution layers, a global mean pooling layer aggregates node features into a cohesive graph-level representation. This representation is then processed through a fully connected layer, which outputs class probabilities using a logarithmic softmax function. For GoG models, we utilize publicly available code from the Github repositories of the original studies. For GoGNN and DVGGA, we adapt the original code from edge prediction to node classification tasks on the global graph.

**Hyperparameters.** For individual GNN models, we configure each layer with a dimension of 16, a dropout rate of 0, a learning rate of 0.01, and set the number of training epochs to 50. Cross-entropy serves as the loss function. For GoG-based models using a single GCN model as the backbone, we ensure that the dimensions and dropout rates are consistent with those of the individual GNN models. To fine-tune additional hyperparameters, we experiment with various settings listed in Table 6 to achieve optimal performance.

Table 6: GoG models parameter settings.

| Model | Parameter | Values |
|---|---|---|
| SEAL | First dense neurons | 16, 32, 64 |
| | Second gcn dimensions | 8, 16 |
| | Number of epochs | 50, 100, 150 |
| | Weight | 0, 0.001, 0.00001 |
| GOGNN | Nhid | 32, 64, 128 |
| | Number of epochs | 50, 100, 150 |
| | Pooling rate | 0.4, 0.5, 0.6 |
| DVGGA | Vgae hidden dimensions | 8, 16, 32 |
| | Number of epochs | 50, 100, 150 |

### E.2 Anomaly Detection

**Models.** We test two groups of models: (1) 4 models for multivariate anomaly detection, and (2) 5 models for the graph anomaly detection.

Group (1) includes probabilistic-based and outlier ensembles methods designed for multivariate anomaly detection:

- *Copula-Based Outlier Detection (COPOD)* [54] is a probabilistic model that leverages the advantages of copulas for outlier detection. It does not assume a normal distribution of data, making it robust and effective in identifying outliers in various datasets with complex distributions.
- *Isolation Forest (IForest)* [55] utilizes a decision tree structure to isolate outliers by randomly selecting features and split values between the feature's maximum and minimum. Its efficiency and scalability make it well-suited for large datasets.
- *Deep Isolation Forest (DIF)* [56] extends the traditional isolation forest by incorporating deep learning techniques to enhance its capability to handle high-dimensional and complex structured data.
- *Variational Autoencoder (VAE)* [57] is a generative model that uses a neural network architecture to model data distributions and encode data into a latent space. It is widely used for anomaly detection by reconstructing inputs and measuring reconstruction errors to identify anomalies.

Group (2) includes anomaly detection methods on graphs, primarily utilizing GNN combined with Autoencoder techniques:

- *Graph Autoencoder (GAE)* [58] employs a graph convolutional network to encode the graph structure into a latent space, then reconstructs the graph to identify anomalies by measuring reconstruction loss.
- *Detection of Outliers in Network Data (DONE)* [59] integrates graph structural information with node feature information to detect anomalous nodes effectively within graph data.
- *Deep Anomaly Detection on Attributed Networks (DOMINANT)* [60] uses a deep autoencoder model adapted to graph data, enhancing the ability to capture non-linearities and complex patterns in the data, which helps in identifying both global and local anomalies in graphs.
- *Anomaly Detection with Autoencoder (AnomalyDAE)* [61] is an autoencoder-based model that particularly focuses on detecting anomalies in dynamic graphs by learning a representation that captures both the graph structure and changes over time.
- *Contrastive Learning for Anomaly Detection (CoLA)* [62] utilizes contrastive learning to differentiate between normal and abnormal nodes, leveraging the discriminative power of contrastive loss to enhance anomaly detection performance in graph settings.

**Hyperparameters.** We test on the following hyperparameters in Table 7 and select the best setting with superior performance.

Table 7: Models of anomaly detection parameter settings. $n$ represents the number of features.

| Model | Parameter | Values |
| --- | --- | --- |
| COPOD | Contamination | 0.01 to 0.1 (linear space) |
| Isolation Forest | Number of estimators
Maximum samples | 100, 200
256, 512 |
| DIF | Contamination | 0.01 to 0.05 (linear space) |
| VAE | Encoder neurons
Decoder neurons
Contamination | $n/4$, $n/2$, $\min(20, n)$
$n/4$, $n/2$, $\min(20, n)$
0.1 to 0.3 (linear space) |
| DOMINANT, DONE, GAE, AnomalyDAE, CoLA | Hidden dimensions
Learning rate
Number of epochs | 16, 32, 64
0.01, 0.005, 0.1
50, 100, 150 |

# F   Global Link Prediction

Link prediction is an essential task in graph learning, widely applied in recommendation systems [74] and social media analysis [75]. In the context of blockchain analysis, predicting interactions between tokens is essential for forecasting future market behaviors. This section focuses on global edge prediction, specifically aiming to forecast interactions for newly launched tokens using information from existing tokens.

**Models.** We compare two groups of models based on the previous section subsection 5.1. The first group consists of traditional Graph Neural Network (GNN) models applied to global token graphs. The second group includes Graphs of Graphs (GoG) models, which leverage the hierarchical structure of token-to-token interactions. We provide a detailed comparison of performance metrics to substantiate our claims regarding the effectiveness of these models.

**Settings.** Our analysis focuses on the most recent tokens launched within the past year. We divided global token-token interactions into training and test sets, following an 80/20 ratio based on the tokens' launch times. Node degree serves as the primary feature for local graph embeddings, consistent with our approach in the classification task. We evaluate model performance using accuracy and AUC, supplemented by precision and recall to provide a comprehensive assessment.

**Results.** The performance of global edge prediction methods across three blockchains is summarized in Table 8. As shown, GoG models do not consistently outperform individual GNN models, particularly on the BSC dataset. One potential reason for these results is that the node degree, used as a node feature in this experiment, may not be as effective for predicting global edges as it is for

Table 8: Edge prediction performance by blockchain.

| Model | Ethereum | | Polygon | | BSC | |
|---|---|---|---|---|---|---|
| | Accuracy | AUC | Accuracy | AUC | Accuracy | AUC |
| GCN | $58.07_{\pm0.36}$ | $62.02_{\pm0.23}$ | $59.64_{\pm1.71}$ | $66.92_{\pm5.37}$ | $66.73_{\pm3.12}$ | $72.87_{\pm3.42}$ |
| GAT | $50.80_{\pm0.43}$ | $54.50_{\pm2.43}$ | $50.70_{\pm2.07}$ | $54.64_{\pm4.47}$ | $52.82_{\pm0.77}$ | $53.62_{\pm2.86}$ |
| GIN | $56.48_{\pm1.61}$ | $56.36_{\pm1.77}$ | $59.03_{\pm3.47}$ | $58.17_{\pm4.33}$ | $59.98_{\pm2.61}$ | $63.57_{\pm3.48}$ |
| ResidualGCN | $50.31_{\pm0.37}$ | $50.66_{\pm0.54}$ | $49.91_{\pm0.08}$ | $49.92_{\pm0.10}$ | $50.41_{\pm0.43}$ | $50.74_{\pm0.94}$ |
| GraphSage | $50.92_{\pm1.03}$ | $53.67_{\pm2.11}$ | $56.63_{\pm8.88}$ | $60.17_{\pm12.83}$ | $\mathbf{71.02} \pm 0.05$ | $\mathbf{78.07} \pm 1.08$ |
| SEAL | $57.09_{\pm1.64}$ | $64.74_{\pm4.83}$ | $56.98_{\pm4.93}$ | $64.62_{\pm10.34}$ | $56.52_{\pm4.62}$ | $58.05_{\pm6.04}$ |
| GoGNN | $\mathbf{66.94} \pm 2.08$ | $\mathbf{72.04} \pm 2.41$ | $57.10_{\pm5.21}$ | $56.72_{\pm4.75}$ | $58.99_{\pm2.77}$ | $66.25_{\pm1.84}$ |
| DVGGA | $50.40_{\pm1.79}$ | $62.93_{\pm1.73}$ | $\mathbf{72.38} \pm 1.36$ | $\mathbf{76.00} \pm 0.32$ | $63.63_{\pm4.94}$ | $69.11_{\pm3.95}$ |

classification tasks. This suggests that further exploration of edge feature engineering could enhance the predictive capabilities of GoG models for token-token interactions.

Additionally, the dynamic nature of blockchain networks presents opportunities to monitor and predict future token-token interactions, which could forecast significant market trends. However, most current GoG models are not designed with dynamic algorithms [19, 20], highlighting both challenges and potential areas for further research. We recommend future work to explore the integration of dynamic features and more sophisticated edge feature engineering to improve prediction accuracy. In summary, our findings indicate that while GoG models show promise, there is a need for further refinement and exploration of features to enhance their predictive performance in the context of blockchain networks.

## G   Multi-Class Graph Classification - Temporal Split

In this section, we present additional experiments that focus on predicting the class label of younger tokens using information derived from older tokens. To simulate a realistic scenario where future tokens are classified based on historical data, we implement a temporal split of the dataset. Specifically, we divide the tokens into training and test sets following an 80/20 ratio based on their first transaction timestamps. This approach enables evaluation of the model's performance within a time-sensitive context, which is crucial for applications in dynamic environments like blockchain.

The experimental settings align with those described in subsection 5.1. The results of these experiments are summarized in Table 9, which provides a comparative analysis of classification performance across different models and blockchain platforms.

Upon comparing these results with those presented in Table 4, we observe that the performance for Ethereum and BNB shows only slight differences regardless of the node-splitting method employed. However, for Polygon, we note a significant deterioration in performance. This discrepancy may be due to Polygon's status as the fastest of the major Ethereum-based chains [76], leading to varying transaction patterns across different time periods. These findings suggest that while our methods demonstrate competitive performance, further investigation is warranted to understand the underlying factors affecting classification accuracy across different blockchains.

## H   Graph Anomaly Detection with Deepwalk Embeddings

In this section, we present an effective method for representing token graphs in anomaly detection tasks by employing the DeepWalk algorithm [65]. DeepWalk is well-known for generating robust graph embeddings through the simulation of random walks. This approach captures the network topology and provides a nuanced representation of graph structures.

We configured DeepWalk with a walk length of 20 and performed 40 walks per node on each token transaction graph. This configuration strikes a balance between the depth and breadth of neighborhood exploration, ensuring that the embeddings accurately capture the structural and contextual nuances of the token graphs. We then aggregated these node embeddings into a unified graph-level representation by computing their mean, resulting in an embedding of 32 dimensions for each graph.

Table 9: 3-class and 5-class classification performance by blockchain (node split by time).

| Model | Ethereum | | Polygon | | BSC | |
|---|---|---|---|---|---|---|
| | F1-macro | F1-micro | F1-macro | F1-micro | F1-macro | F1-micro |
| 3-Class Classification | | | | | | |
| GCN | $60.16_{\pm5.60}$ | $87.70_{\pm0.83}$ | $22.37_{\pm0.57}$ | $48.22_{\pm0.67}$ | $50.01_{\pm5.27}$ | $57.39_{\pm3.78}$ |
| GAT | $57.50_{\pm6.25}$ | $87.33_{\pm1.16}$ | $26.00_{\pm2.67}$ | $48.91_{\pm1.02}$ | $51.15_{\pm6.52}$ | $59.48_{\pm5.58}$ |
| GIN | $60.38_{\pm5.76}$ | $87.68_{\pm0.94}$ | $21.74_{\pm1.21}$ | $48.03_{\pm0.63}$ | $42.56_{\pm2.73}$ | $56.59_{\pm3.65}$ |
| ResidualGCN | $40.62_{\pm8.06}$ | $83.83_{\pm1.41}$ | $22.86_{\pm1.02}$ | $48.24_{\pm0.55}$ | $48.09_{\pm5.30}$ | $60.13_{\pm2.95}$ |
| GraphSage | $61.71_{\pm6.27}$ | $88.25_{\pm0.97}$ | $24.91_{\pm1.87}$ | $48.72_{\pm0.55}$ | $53.86_{\pm6.99}$ | $62.16_{\pm4.28}$ |
| SEAL | $\mathbf{67.42}_{\pm1.05}$ | $\mathbf{88.72}_{\pm0.33}$ | $27.20_{\pm1.81}$ | $\mathbf{49.37}_{\pm0.59}$ | $55.14_{\pm5.62}$ | $\mathbf{64.03}_{\pm3.82}$ |
| GoGNN | $66.10_{\pm1.98}$ | $88.28_{\pm0.80}$ | $\mathbf{30.85}_{\pm2.32}$ | $44.75_{\pm4.09}$ | $\mathbf{61.80}_{\pm0.50}$ | $62.17_{\pm0.33}$ |
| DVGGA | $53.80_{\pm1.98}$ | $75.60_{\pm7.67}$ | $28.22_{\pm1.44}$ | $41.52_{\pm0.98}$ | $24.03_{\pm13.78}$ | $35.37_{\pm15.33}$ |
| 5-Class Classification | | | | | | |
| GCN | $38.75_{\pm5.44}$ | $85.18_{\pm0.93}$ | $12.11_{\pm0.53}$ | $41.16_{\pm0.95}$ | $26.76_{\pm3.74}$ | $47.21_{\pm4.33}$ |
| GAT | $37.02_{\pm5.64}$ | $85.24_{\pm1.07}$ | $\mathbf{16.63}_{\pm3.04}$ | $42.10_{\pm2.27}$ | $28.43_{\pm4.08}$ | $49.37_{\pm5.87}$ |
| GIN | $22.69_{\pm1.43}$ | $80.65_{\pm0.52}$ | $12.15_{\pm0.77}$ | $41.06_{\pm0.76}$ | $22.02_{\pm2.97}$ | $43.48_{\pm5.83}$ |
| ResidualGCN | $41.19_{\pm5.45}$ | $85.00_{\pm1.13}$ | $12.03_{\pm0.58}$ | $41.15_{\pm0.70}$ | $24.38_{\pm4.34}$ | $47.78_{\pm6.34}$ |
| GraphSage | $40.51_{\pm5.82}$ | $86.31_{\pm1.10}$ | $14.97_{\pm1.69}$ | $41.98_{\pm0.69}$ | $27.89_{\pm5.48}$ | $49.06_{\pm6.83}$ |
| SEAL | $\mathbf{48.85}_{\pm0.52}$ | $86.29_{\pm0.27}$ | $15.54_{\pm2.32}$ | $\mathbf{42.41}_{\pm0.15}$ | $\mathbf{30.41}_{\pm1.81}$ | $\mathbf{52.65}_{\pm1.09}$ |
| GoGNN | $45.25_{\pm5.83}$ | $\mathbf{86.36}_{\pm0.76}$ | $14.49_{\pm1.94}$ | $41.77_{\pm0.60}$ | $28.29_{\pm3.51}$ | $52.11_{\pm2.66}$ |
| DVGGA | $25.35_{\pm4.28}$ | $68.96_{\pm16.54}$ | $11.65_{\pm0.01}$ | $41.03_{\pm0.00}$ | $10.91_{\pm2.72}$ | $31.36_{\pm4.46}$ |

Table 10: Graph anomaly detection performance using DeepWalk. We report the ratio of number of non-fraud:fraud case of each data at the top.

| Model | Ethereum (8387: 6022) | | Polygon (2257: 58) | | BNB (6339: 1042) | |
|---|---|---|---|---|---|---|
| | AUC | AP | AUC | AP | AUC | AP |
| COPOD | $50.87_{\pm0.09}$ | $42.57_{\pm0.70}$ | $62.16_{\pm8.3}$ | $3.42_{\pm1.62}$ | $52.47_{\pm0.47}$ | $13.82_{\pm0.91}$ |
| IForest | $50.43_{\pm0.28}$ | $42.69_{\pm1.07}$ | $60.95_{\pm8.7}$ | $3.11_{\pm1.12}$ | $52.14_{\pm1.17}$ | $14.02_{\pm0.85}$ |
| DIF | $50.58_{\pm0.31}$ | $42.10_{\pm0.89}$ | $59.72_{\pm6.85}$ | $2.80_{\pm0.55}$ | $52.16_{\pm0.72}$ | $13.77_{\pm0.91}$ |
| VAE | $50.77_{\pm0.34}$ | $42.87_{\pm0.94}$ | $61.86_{\pm7.98}$ | $3.47_{\pm1.69}$ | $51.69_{\pm1.24}$ | $14.00_{\pm0.81}$ |
| GAE | $51.22_{\pm1.39}$ | $41.44_{\pm0.56}$ | $60.81_{\pm1.25}$ | $\mathbf{5.40}_{\pm2.42}$ | $61.15_{\pm2.67}$ | $\mathbf{24.02}_{\pm1.92}$ |
| DONE | $\mathbf{68.86}_{\pm10.27}$ | $32.89_{\pm5.57}$ | $\mathbf{71.29}_{\pm2.21}$ | $1.63_{\pm0.06}$ | $77.55_{\pm0.13}$ | $8.62_{\pm0.01}$ |
| DOMINANT | $60.92_{\pm4.57}$ | $38.12_{\pm2.63}$ | $67.15_{\pm3.41}$ | $2.43_{\pm1.00}$ | $\mathbf{79.73}_{\pm0.07}$ | $8.42_{\pm0.01}$ |
| AnomalyDAE | $65.14_{\pm3.63}$ | $\mathbf{46.12}_{\pm10.08}$ | $57.90_{\pm3.58}$ | $3.44_{\pm0.31}$ | $52.75_{\pm0.98}$ | $15.67_{\pm0.20}$ |
| CoLA | $50.51_{\pm0.42}$ | $41.90_{\pm0.44}$ | $59.61_{\pm3.94}$ | $2.67_{\pm0.77}$ | $54.87_{\pm0.03}$ | $14.88_{\pm0.56}$ |

The results of our anomaly detection analysis using the DeepWalk algorithm are presented in Table 10. Notably, the GoG models generally outperform multivariate outlier detection methods in our experiments, although this may vary depending on the specific characteristics of each dataset. When comparing the results in Table 7, the superiority of GoG models is evident across all three blockchains when using the DeepWalk algorithm, particularly in scenarios with high fraud rates.

It is important to note that the anomaly detection performance on Polygon remains the poorest among the chains, consistent with previous findings in subsection 5.2. While GoG models benefit from the use of the DeepWalk algorithm, the performance of multivariate outlier detection methods appears to decrease. This suggests that the DeepWalk algorithm significantly enhances the effectiveness of GoG models in identifying anomalies.

# I Details of Compute Resources

We use two machine, one for experiements of inidividual GNN, one for experiements of GoG-based GNN. First, all experiments involving individual GNN models were conducted on machine outfitted

with eight NVIDIA GeForce GPUs, each with a maximum power capacity of 350W and 24,576 MiB of available memory. Second, all experiments utilizing GoG-based GNN models were carried out on the machine equipped with eight NVIDIA A100-SXM4-80GB GPUs. These GPUs, each with a maximum power capacity of 400W and a substantial 81,920 MiB of memory, are specifically chosen for their high performance and large memory capacity, which are ideal for the complex and memory-intensive computations required by GoG-based GNN models.

## J   License

The dataset is released under the Creative Commons Attribution-NonCommercial-ShareAlike (CC BY-NC-SA) license.

## K   Hosting Plan

We choose GitHub as our hosting platform for both code and data due to its ease of use, cost-effectiveness, and scalability. Ensuring easy access to our data is crucial. To facilitate straightforward and reliable data retrieval, we will maintain a curated interface. We are committed to keeping our platform stable and functional, with regular updates and maintenance to ensure our repository remains up-to-date, bug-free, and efficient.

Our project is driven by a commitment to open access. By regularly updating our GitHub repository, we ensure that users have timely access to the latest data. We believe that GitHub's user-friendly environment will provide a dependable and efficient solution for sharing our data with the global community.

