# OpenReview forum: "Multi-Chain Graphs of Graphs: A New Approach to Analyzing Blockchain Datasets"
_NeurIPS.cc/2024/Datasets_and_Benchmarks_Track — NeurIPS 2024 Track Datasets and Benchmarks Poster_

### Official Review · Reviewer_vWPB · 2024-07-20
**three blockchain datasets**

**Rating:** 6
**Confidence:** 4

**Review:**

The paper presents an aggregation scheme for publicly available blockchain data that is focused around 'tokens' as the main entities that have associated temporal transaction graphs, interactions among them, and associated labels for a supervised prediction task. The paper in places remains vague and it remains unclear to me why the proposed learning task is relevant. This is mainly due to missing information about the labels (how are they created, is it expensive to obtain them, how reliable are they, how many tokens on the blockchain are labeled, ...?) and an analysis that does not seem to include the temporal/dynamic nature of the data.
Overall, the creation of the dataset may provide value to the community in the sense that it is (more) easily accessible than the base data. But I belive that in its current state, the paper as a description of the creation process of the dataset is not sufficient for acceptance at this venue, yet.

**Strengths:**

- The paper provides reasonably easy access to an aggregation of blockchain data centered around tokens as the main entities with associated labels.
- The hierarchical nature of the graph of graphs is interesting, while the global graph may be a bit simplistic and is not well-motivated.

**Additional Feedback:**

In places, some sentences feel not completely grammatically correct to me.
E.g., Line 39-42

Line 157 is very imprecise and does not add anything. You may either remove it or give a proper definition of the clustering coefficient.

In Fig. 5a, there is a horizontal empty area. Why is that?

In line 253, a reference seems to be missing

Finally, do you actually provide three independent datasets in the same data format, or is there any kind of overlap (how much) between the Ethereum, Polygon, and BNB datasets, e.g., in tokens, or accounts?

**Clarity:**

The paper implicitly assumes quite some knowledge about blockchain systems. As a result, I had difficulties following it in several places, where concepts appear on the fly when they are used for dataset construction. A brief introductory section introducing involved entities (accounts, tokens, transactions, ...) would be very helpful.  After skimming the supplementary materials, it seems that some description is provided there. A reference in the main paper might do the trick.

Additional descriptions of the semantics ground truth labels are required.

**Correctness:**

The description of the actual acquisition of the data is rather brief. From what can be seen, it seems to be a reasonable approach, while justification for particular cutoff values is not provided. The creation process of 'ground truth' labels remains unclear and needs to be discussed further.

**Documentation:**

Some documentation and links to the experimental code are provided in the appendix. At a quick glance, the code to actually construct the dataset seems not to be provided.
The hosting plan promises data and code availability on GitHub (and potentially Google Drive) as well as 'regular' updates without mentioning any frequency.

**Ethics:**

I see the provided dataset as a postprocessing of publicly available data. In that sense, releasing the dataset creates a higher level of convenience, but does not in itself allow malicious activities that were not possible before.

As previously mentioned, though, I am skeptical about the proposed learning task, as long as no additional discussion regarding the creation process of the labels is provided.

**Limitations:**

The authors address some limitations in section 7 and focus on privacy issues such as tracking individual wallet behavior and surveillance, which in principle is not due to their dataset, but due to the fact that the blockchain data is publicly available.

More severely, they mention that they only extracted a subset of all records with a large number of transactions via publicly available APIs. The description of the selection process is brief and I would, from the text in the paper, be unlikely able to reproduce the dataset. Code seems not to be provided, while the dataset is accessible via Google Drive.

Another limitation that seems not to be addressed is the acquisition of the 'ground truth' labels for the tokens. The paper mentions retrieving them from a service, but how they are created (e.g. manually, or via some automated process) is not discussed. As the labels contain a rather prominent 'fraud' category, this question is a relevant limitation that needs to be addressed. Particularly in conjunction with the privacy issues mentioned before.

**Opportunities For Improvement:**

- The description of the underlying blockchain concepts should be improved. For example, 'accounts' are only introduced when they are mentioned as being the nodes of the local graphs. As the overlap of accounts between local token graphs is used to decide the edge weights in the global graph, it would be important to know more about them. E.g., I would assume that nobody stops a natural or juridical person from creating multiple accounts on the same blockchain. Extending the description of such phenomena could allow us to understand the limitations and possibilities of this dataset.
- The authors mention that their graph dataset is dynamic, but do not go into detail in their description. I assume that the dynamic nature refers to time stamps on the transaction edges in the local graphs. I assume that tokens have at least a creation date that could be useful, but this is not mentioned, as far as I can see.
The analysis in section 4 and the experiments in section 5 seem to not take this dynamic nature into account. For example, to improve the practical use of the experiments in section 5, a temporal split might be in order, i.e., predicting the class label of younger tokens based on information about old tokens, etc.
- The global graph seems to be static, as the Jaccard similarity seems to be defined over the static sets of accounts in the local graphs. Can this be made dynamic, using the dynamic transaction information in the local graphs?

**Relation To Prior Work:**

The paper cites several other datasets. I am not aware of any relevant missing literature, but I am not an expert in ML on blockchains.

**Summary And Contributions:**

The authors present three blockchain hierarchical graph datasets with two levels. Local graphs are constructed for each 'token' that is being sold on a blockchain. Nodes of the local graphs are accounts and edges correspond to transactions regarding the token between accounts. In the global graph, these 'token' graphs are connected by an edge if their local graphs share common accounts.

Tokens have labels (e.g. Fraud) and the authors present classification and anomaly detection results on their dataset.

---

> ### Author Rebuttal · Authors · 2024-08-17
>
> **Q1: The description of the underlying blockchain concepts should be improved...**
>
> A1: Thank you for your feedback. In Appendix A, we provide a more detailed description of key blockchain concepts. We would move some key concepts description to the main paper as a background section. Thank you for helping us strengthen our work!
>
> An account on an Ethereum Virtual Machine (EVM) blockchain is an entity capable of sending transactions and interacting with smart contracts. Each account has a unique address, maintains a balance, and is controlled by a private key. Thank you for pointing out that there are no restrictions preventing a natural or juridical person from creating multiple accounts on the same blockchain. This phenomenon indeed introduces certain challenges to our dataset. Multiple accounts can be owned by a single entity, potentially skewing the interaction patterns and connectivity metrics within local graphs.
>
>
> **Q2: The authors mention that their graph dataset is ...**
>
> A2: Thank you for your valuable feedback. Your assumptions are correct. The timestamps of the transactions indicate when the transactions happen. Timestamp of the first transaction of each token can be regarded when it becomes active on chain. We would add it to Section 3.2.1 when we introduced the local graph construction.
>
> We have conducted additional experiments predicting the class label of younger tokens using the information about old tokens. In specific, we divide tokens into training and test sets, following a 80/20 ratio based on its first transaction timestamp. Other settings are the same as our experiments described in Section 5.1. The results of these experiments are included in the pdf file attached.
>
> When comparing these results to those in Table 4 of Section 5.1, we observe that for Ethereum and BNB, the performance shows slight differences from the results in Table 4 (where the training and test sets were randomly split). However, for Polygon, the performance deteriorates significantly. A possible explanation is that Polygon is currently the fastest among the major Ethereum-based chains [1], and transaction patterns may vary across different time periods.
>
> Reference:
> [1] https://www.binance.com/en/square/post/8936643909185.
>
>
> **Q3: The global graph seems to be static, as the Jaccard similarity seems...**
>
> A3: Thank you for your question. Indeed, global edges in our graph can dynamically adapt to changes in the local transaction information. As detailed in Appendix D, we analyze the yearly temporal evolution of the global graph to understand how blockchain networks adapt over time. For example, for the past three years, on average, the increase of number of nodes on global graphs reaches 42.49% for Ethereum, 33.08% for Polygon, and 65.18% for BnB, indicating significant changes in the dataset.
>
> Moreover, using this dynamic transaction information would also allows us to monitor and predict significant trends, such as future token-token interactions, potentially forecasting future market behaviors. However, we found out that one limitation of most existing GoG models is that they do not design dynamic algorithms [1, 2]. Thus, we believe this setting will bring new opportunities and challenges to the graph research.
>
> [1] Li, J., Rong, Y., Cheng, H., Meng, H., Huang, W., & Huang, J. (2019, May). Semi-supervised graph classification: A hierarchical graph perspective. In `The World Wide Web Conference` (pp. 972-982).
>
> [2] Wang, H., Lian, D., Zhang, Y., Qin, L., & Lin, X. (2020). Gognn: Graph of graphs neural network for predicting structured entity interactions. `arXiv preprint arXiv:2005.05537.`

---

> ### Author Rebuttal · Authors · 2024-08-17
>
> **Limitation: The authors address some limitations...**
>
> A: Thank you for your feedback. We agree with your comments that the privacy issues such as tracking individual wallet behavior and surveillance is mainly due to the fact of blockchain data itself.
>
> For the dataset and code, our code is documented in GitHub [https://anonymous.4open.science/r/Graph-of-graph-dataset-B2E3/README.md] as started in Appendix I. We document how we build the graph in the code. We will move it to the paper abstract in the next version of the paper.
>
> For the ground truth of the labels, in Section 3.1 of our paper, we initially described our collection of labels sourced from three leading blockchain explorers: Etherscan [1], Polygonscan [2], and Bscscan [3]. These platforms are renowned for their extensive use and reliable data within the blockchain community. In response to your concerns, we have enhanced our description to include our systematic process for label verification and class grouping, which we detail below (this part is to be added to Section 3.1):
>
> For fraud cases, we labelled the tokens which are described to be suspicious phish/hack tokens by these explorer. Those tokens include various kinds of spammed tokens, such as those have been spammed to many users or those who pretend to be some famous tokens, such as fake USDT. For example, for token 0x8f1a7720d0798d50a01111b15f8c48d43cbc96af on Ethereum, it is highlighted as “This token is reported to have been spammed to many users. Please exercise caution when interacting with it.” on its token page on Etherscan [https://etherscan.io/token/0x8f1a7720d0798d50a01111b15f8c48d43cbc96af]. In total, 7,978 tokens were identified as suspicious, representing about 32.8% of the dataset.
>
> For other classes, we labelled the tokens using the category tags given by the explorer. For example, for token 0xa808b22ffd2c472ad1278088f16d4010e6a54d5f on Ethereum, it has tag "finance" on Etherscan which indicate it's a token related to finance.[https://etherscan.io/token/0xa808b22ffd2c472ad1278088f16d4010e6a54d5f]. The most popular classes in the dataset include DeFi tokens, which are related to decentralized finance products; MEME tokens, often inspired by internet memes and characters; and Gaming tokens, which are associated with electronic gaming. We do not manually class any category.
>
> Reference
> [1] Etherscan. https://etherscan.io/.
> [2] Polygonscan. https://polygonscan.com/.
> [3] Bscscan. https://bscscan.com/.
>
>
>
> **Documentation: Some documentation and links to the experimental code...**
>
> A: Thank you for your feedback. Please check our data collection code here [https://drive.google.com/file/d/14eUvQ_7PPpjzuvnobBE0V-mgajLoBrd4/view?usp=sharing]. It follows the API documentation of the blockchain explorer. We would add it to the github repo. For the data and code availability, we will maintain and update it monthly.
>
>
> **Additional Feedback:**
>
> **Q1: In places, some sentences feel not completely grammatically correct to me. E.g., Line 39-42**
>
> A1: Thank you for your feedback. We revised the sentence structure to ensure clarity and precision:
>
> *"Second, it integrates token-token interactions across multiple blockchain platforms. Specifically, our dataset includes 268,282,924 transactions conducted by 18,600,142 cryptocurrency addresses, covering the transaction history of 24,316 tokens on three main EVM-chains: Ethereum, Polygon, and BNB."*
>
>
>
> **Q2: Line 157 is very imprecise and does not add anything. You may either remove it or give a proper definition of the clustering coefficient.**
>
> A2: Thank you for your feedback. In appendix C, we introduced the definition of the clustering coefficient. We will add these lines to make it clearer:
>
> *"The clustering coefficient indicates how closely nodes in a graph cluster, reflecting the formation of tight-knit groups or collusive clusters. In the token transfer graph, a higher average clustering coefficient suggests the presence of prevalent cliques or active trading communities."*
>
>
>
> **Q3: In Fig. 5a, there is a horizontal empty area. Why is that?**
>
> A3: Thank you for your question. The empty area in Figure 5a corresponds to a range of transaction counts where few tokens are present. Specifically, we conducted an analysis of the distribution of the y-values, which represent the number of transactions for the tokens. We found that very few tokens have between 150 and 250 transactions. Since the y-values are plotted on a log scale, this sparse distribution creates a noticeable empty area in the figure.
>
>
>
> **Q4: In line 253, a reference seems to be missing**
> A4: Thank you for your comment. The missing references are [61, 62] in our reference. We added the two missing references, which are [61, 62] in our reference.
>
> Reference:
> [61] Yue Zhao, Zain Nasrullah, and Zheng Li. Pyod: A python toolbox for scalable outlier detection. *Journal of Machine Learning Research*, 20(96):1–7, 2019.
>
> [62] Kay Liu, Yingtong Dou, Xueying Ding, Xiyang Hu, Ruitong Zhang, Hao Peng, Lichao Sun, and Philip S. Yu. PyGOD: A Python library for graph outlier detection. *Journal of Machine Learning Research*, 25(141):1–9, 2024.
>
>
>
> **Q5: Finally, do you actually provide three independent datasets in the same data format, or is there any kind of overlap (how much) between the Ethereum, Polygon, and BNB datasets, e.g., in tokens, or accounts?**
>
> A5: Thank you for your question. We provide three independent datasets corresponding to the Ethereum, Polygon, and BNB blockchain platforms. There is no overlap in accounts or tokens among these datasets, as each set is specific to its respective platform.

---

> > ### Comment · Reviewer_vWPB · 2024-08-21
> >
> > Thank you for your detailed reply.
> > While most of your replies are sufficient, I am not yet convinced by your argument on the limitations.
> >
> > From your description, I now gather that the labels for the tokens are completely provided by the blockchain explorers. However, there is not yet any information available on how these explorers come up with these labels. This poses two challenges for the present paper:
> > 1) Practical Value: If I can access the label for any token via a simple API call to etherscan et al., why would I need to train a model to predict it? I.e., what is the use case of a hypothetical model predicting these labels?
> > 2) Ethical Implications: If the algorithm of e.g. Etherscan wrongly classifies a token as fraud, any model would -- by training -- try to mimic this behavior. Given that 'fraud' may have severe legal implications for people accused of committing it, this is an important point that needs to be addressed in more detail.

---

> > > ### Author Response · Authors · 2024-08-22
> > >
> > > Thank you for your reply. The blockchain explorers come up with these labels from the blockchain and the Internet [1].
> > >
> > > Regarding practical value, the purpose of our model goes beyond merely replicating existing blockchain explorer labels. The idea is to enhance predictive capabilities in scenarios where labels might be incomplete or unavailable. In fact, only a small portion of existing tokens have been labeled by these blockchain explorers (for example, on Ethereum, there are more than 900,000 ERC20 tokens in total), and new tokens are launched every day that may not yet be classified. Our model could predict these labels faster and offer more detailed insights.
> > >
> > > Regarding ethical implications, we agree this is an important point and have taken steps to mitigate this concern. First, following previous related works on blockchain [2, 3], we believe that the blockchain explorers we selected are generally considered trustworthy. Second, we have verified the tokens labeled as fraudulent on these blockchain explorers. We found that when blockchain explorers label a token, they not only provide the label but also offer valid justifications. For instance, token 0xbd6323a83b613f668687014e8a5852079494fb68 is labeled as "Phish/Hack" on Ethereum, with an explanation stating, “We have received reports that this is not an official token issued by BlackRock and is not associated with the brand. Please treat it with caution.” This token falsely presents itself as BlackRockTradingCurrency [https://etherscan.io/token/0x8f1a7720d0798d50a01111b15f8c48d43cbc96af]. We believe these highlights will help enhance the reliability of the labels. Third, while we acknowledge that some errors may occur, e.g. Etherscan wrongly classifies a token as fraud, we believe the cost of missing fraudulent tokens is far greater. As of 2023, crypto losses have reached billions of US dollars [4]. Therefore, while some errors may occur in the initial steps, it is worthwhile to conduct further checks on suspicious labels due to the high cost of missing fraudulent tokens.
> > >
> > > Moreover, in the limitation section in our paper, we will add that our model is intended to augment, not replace, human judgment. Predictions would serve as suggestions that require further scrutiny, especially in high-stakes contexts such as fraud detection.
> > >
> > > Reference:
> > > [1] https://info.etherscan.com/public-name-tags-labels/
> > > [2] Yuan, Q., Huang, B., Zhang, J., Wu, J., Zhang, H., & Zhang, X. (2020, October). Detecting phishing scams on ethereum based on transaction records. In 2020 IEEE international symposium on circuits and systems (ISCAS) (pp. 1-5). IEEE.
> > > [3] Li, S., Gou, G., Liu, C., Hou, C., Li, Z., & Xiong, G. (2022, April). TTAGN: Temporal transaction aggregation graph network for ethereum phishing scams detection. In Proceedings of the ACM Web Conference 2022 (pp. 661-669).
> > > [4] https://de.fi/blog/de-fi-rekt-report-crypto-losses-reach-1-95b-in-2023

---

> > > ### Author Response · Authors · 2024-08-25
> > >
> > > Dear Reviewer vWPB,
> > >
> > > Thank you for your detailed comments on our work. Your feedback helped us improve our paper's quality. As the discussion period is nearing its end, could you please provide a response to our rebuttal? If there are any aspects that are unclear or if you would like further explanation, we are more than happy to clarify. Thank you very much for your time and consideration.
> > >
> > > Best regards,
> > > The Authors

---

> > > ### Author Response · Authors · 2024-08-28
> > > **Gentle Reminder to Reviewer vWPB**
> > >
> > > Dear Reviewer vWPB,
> > >
> > > We sincerely appreciate the time and effort you've dedicated to reviewing our paper. In response to your feedback, we summarize and add more important points to our limitation section in the paper as below:
> > >
> > > `Our dataset faces several limitations. First, the lack of restrictions on creating multiple accounts on the same blockchain allows a single entity to own multiple accounts. This poses a risk of distorting interaction patterns and connectivity metrics within our graphs of graphs dataset. Second, while our model is designed to enhance predictive capabilities in scenarios where labels are incomplete, it is limited by the fact that only a small fraction of existing tokens have been labeled by blockchain explorers, and new tokens are frequently launched without immediate classification. Although we rely on trusted blockchain explorers for labeling, there is always a risk of misclassification, particularly concerning fraudulent tokens. Therefore, our model is intended to augment, not replace, human judgment, with predictions serving as suggestions that require further scrutiny. Third, the public nature of blockchain data introduces privacy concerns, as our dataset, which links transactions to wallet addresses, may enable the tracking of individual behaviors, potentially leading to targeted advertising or surveillance. `
> > >
> > > As the discussion period concluding in approximately three days, could you please provide a response to our rebuttal? If there are any aspects that are unclear or if you would like further explanation, we are more than happy to clarify. Thank you for your time and consideration!
> > >
> > > Best regards,
> > >
> > > The Authors

---

> > > > ### Comment · Reviewer_vWPB · 2024-08-29
> > > >
> > > > Thank you for your extensive reply and for presenting the updated version of the limitations section. I am also sorry for my late reply. I know that this phase can be stressful.
> > > > I will increased my score to reflect that the authors have addressed most of my concerns sufficiently and I am now confident that the provided dataset can be a useful resource for the graph ml / blockchain community, when the promised changes are made to the paper for the camera ready version.

---

> > > > > ### Author Response · Authors · 2024-08-29
> > > > >
> > > > > Dear Reviewer vWPB,
> > > > >
> > > > > Thank you for your insightful feedback and for the improved rating. We are pleased to have addressed your concerns effectively and are grateful for your valuable suggestions, which we will incorporate into our paper.
> > > > >
> > > > > Thank you once again for your support!
> > > > >
> > > > > Best regards,
> > > > >
> > > > > The Authors

---

### Official Review · Reviewer_kLuX · 2024-07-24
**Dataset construction is reasonable but the motivation & analysis is a bit unclear**

**Rating:** 5
**Confidence:** 5
**Correctness:** The claims made in the submission app…
**Clarity:** The paper is well written.

**Review:**

GoG systems are subjects of great interest, and curating datasets for them could be potentially impactful in the context of this venue. The size of cryptocurrency platforms and the graphs that are induced are considerable, and I'd like to commend the authors for doing a good job in curating it and processing it into a format that can then be applied to various graph representation learning algorithms. My main concern with the work is that it appears more expositional than insightful: the majority of the work is spent detailing the collection process, properties of the resulting GoG, and some analysis & experiments, without fully explaining 1. why GoG of blockchains is important and 2. what the main motivations and insights, novel or not, from analyzing them are, in the context of "typical" GoG systems in chemistry and information retrieval. This also makes the experiments run on the dataset, though well documented and explained, appear superficially motivated: what are the categories that one is predicting in terms of local graph classification? How are fraudulent tokens defined? Hence, my recommendation is borderline towards rejection.

**Strengths:**

* This is a fairly sizeable dataset and of certain value to the graph representation learning community
* For the experiments themselves, a wide variety of models are trained, and the process is described in fine detail
* Overall, the paper is fairly easy to follow

**Additional Feedback:**

No additional feedback.

**Documentation:**

The dataset construction process is reasonably well documented and there appears to be enough details to replicate the experimental results.

**Ethics:**

I do not foresee any ethical concerns. It is important, though, to point out that the authors "assume full responsibility for any liability or infringement of third-party rights that may come up from the use of our data", as indicated in the supplementary materials.

**Limitations:**

The authors appear to have adequately addressed the limitations and potential negative societal impact of their works.

**Opportunities For Improvement:**

* Minor misprints (e.g. L253, [], Figure 4 could be made to be black & white friendly)
* I believe the audience of this venue could be benefited by a more succinct overview of why blockchain data, especially GoG, are of great interest and importance
* Design experiments that are more clearly motivated - e.g. prediction targets that capture token-token interactions
* Provide more clarity on experimental design, especially how labels are derived, and what each category stands for.
* Explain the pros and cons of taking a snapshot of the state of blockchain

**Relation To Prior Work:**

The introduction is fairly well written, with some discussion with respect to previous works.

**Summary And Contributions:**

The work takes snapshots of popular cryptocurrency platforms (that are themselves blockchains) to produce a large-scale Graph of Graphs (GoG) system, where the local graph represents transactions within a particular cryptocurrency token, and the global graph represents relationships between the tokens. The dataset collection process is then explained, as well as experiments run on top of the dataset consisting of graph classification and anomaly detection.

---

> ### Author Rebuttal · Authors · 2024-08-17
>
> **Q1: Minor misprints (e.g. L253, [], Figure 4 could be made to be black & white friendly)**
>
> A1: Thank you for your feedback. We added the two missing references, which are [61, 62] in our reference.
> [61] Yue Zhao, Zain Nasrullah, and Zheng Li. Pyod: A python toolbox for scalable outlier detection. *Journal of Machine Learning Research*, 20(96):1–7, 2019.
>
> [62] Kay Liu, Yingtong Dou, Xueying Ding, Xiyang Hu, Ruitong Zhang, Hao Peng, Lichao Sun, and Philip S. Yu. PyGOD: A Python library for graph outlier detection. *Journal of Machine Learning Research*, 25(141):1–9, 2024.
>
> For the Figure 4, we have modified to make it black and white friendly. Please refer to the attached pdf. Thank you!
>
>
> **Q2: I believe the audience of this venue could be benefited by a more succinct overview of why blockchain data, especially GoG, are of great interest and importance
> Related questions in the review: 1. why GoG of blockchains is important and 2. what the main motivations and insights, novel or not, from analyzing them are, in the context of "typical" GoG systems in chemistry and information retrieval.**
>
> A2: Thank you for your questions and suggestions, as well as the relevant points raised in the "Review" section. Blockchain data, with its complexity and rich structural information, presents unique challenges and opportunities for advanced machine learning techniques. On the blockchain, there is a wide variety of digital tokens representing diverse assets, such as DeFi tokens related to decentralized finance products and protocol tokens used for communication between connected computers. While these tokens are distinct, they are also interconnected, as they are implemented on the same blockchain and can interact with the same user groups. However, most previous studies either treat all tokens collectively or analyze individual tokens in isolation, without considering the relationships between them [1, 2]. Applying GoG to blockchain data is particularly promising, as it facilitates a deeper understanding of multi-level interactions among tokens in the blockchain ecosystems. This approach provides a comprehensive framework for analyzing how individual transactions shape token behaviors and how these behaviors are interconnected across different blockchain platforms.
>
> Compared to other "typical" GoG systems, such as those in chemistry and information retrieval, blockchain GoGs involve large-scale local graphs, dense global graph structures, and dynamic real-life temporal edges. These characteristics are detailed and compared in Table 1. For example, in our datasets, the average number of nodes in the local graphs exceeds 1,000, far surpassing the typical scale of local graphs in GoG datasets from chemistry or drug research, which usually contain fewer than 100 nodes.
>
> Because of these differences, most existing GoG models are designed for static, smaller local graphs, which contrasts with the large-scale and dynamic nature of blockchain graphs, resulting in significant computational demands in terms of time and memory. Moreover, our experiments, discussed in Section 5, demonstrate that while existing GoG models generally outperform individual GNN models, they also present limitations on the blockchain graphs. For instance, as the complexity of the classification task increases with the addition of minor classes, the advantages of GoG models diminish. Moreover, these models do not consistently provide advantages in anomaly detection tasks.
>
> Therefore, we believe this dataset and modeling approach bring new opportunities and challenges to the fields of blockchain and graph research, suggesting promising areas for future development.
>
> [1] Federico Cernera, Massimo La Morgia, Alessandro Mei, and Francesco Sassi. Token spammers, rug pulls, and sniper bots: An analysis of the ecosystem of tokens in ethereum and in the binance smart chain. In *32nd USENIX Security Symposium (USENIX Security 23)*, pages 3349–3366, 2023.348
>
> [2] Zhen Zhang, Bingqiao Luo, Shengliang Lu, and Bingsheng He. Live graph lab: Towards open, dynamic and real transaction graphs with nft. *Advances in Neural Information Processing Systems*, 36, 2023
>
>
> **Q3: Design experiments that are more clearly motivated - e.g. prediction targets that capture token-token interactions**
>
> A3: Thank you for your suggestion. We agree this would be very meaningful task and our dataset can support such task. We have conducted experiment for global edge prediction, focusing on the most current tokens that launched in the last year. In this experiment, we divide global edges (token-token) into training and test sets, following a 80/20 ratio based on the launch time of the tokens. In other words, we try to predict the token-token interactions for new launched tokens use the information of the existing ones. For each dataset, we compare the performance of GNN models applied on the global graphs and the GoG models. We show our results in the attached pdf.
>
> As shown in the table, GoG models do not outperform individual GNN models in this experiment. One possible reason is that we currently use node degree as the node feature for local graph embeddings, following the same settings as in the classification task, which may not be effective for global edge prediction. This suggests potential areas for improvement in GoG models, such as exploring more advanced edge feature engineering to enhance predictive performance on token-token interactions.

---

> ### Author Rebuttal · Authors · 2024-08-17
>
> **Q4: Provide more clarity on experimental design, especially how labels are derived, and what each category stands for.**
>
> A4: Thank you for your valuable feedback. We recognize the importance of transparency in data source and labeling for maintaining the quality and trustworthiness of research in blockchain research.
>
> In Section 3.1 of our paper, we initially described our collection of labels sourced from three leading blockchain explorers: Etherscan [1], Polygonscan [2], and Bscscan [3]. These platforms are renowned for their extensive use and reliable data within the blockchain community. In response to your concerns, we have enhanced our description to include our systematic process for label verification and class grouping, which we detail below (this part is to be added to Section 3.1):
>
> For fraud cases, we labelled the tokens which are described to be suspicious phish/hack tokens by these explorer. Those tokens include various kinds of spammed tokens, such as those have been spammed to many users or those who pretend to be some famous tokens, such as fake USDT. For example, for token 0x8f1a7720d0798d50a01111b15f8c48d43cbc96af on Ethereum, it is highlighted as “This token is reported to have been spammed to many users. Please exercise caution when interacting with it.” on its token page on Etherscan [https://etherscan.io/token/0x8f1a7720d0798d50a01111b15f8c48d43cbc96af]. In total, 7,978 tokens were identified as suspicious, representing about 32.8% of the dataset.
>
> For other classes, we labelled the tokens using the category tags given by the explorer. For example, for token 0xa808b22ffd2c472ad1278088f16d4010e6a54d5f on Ethereum, it has tag "finance" on Etherscan which indicate it's a token related to finance.[https://etherscan.io/token/0xa808b22ffd2c472ad1278088f16d4010e6a54d5f]. The most popular classes in the dataset include DeFi tokens, which are related to decentralized finance products; MEME tokens, often inspired by internet memes and characters; and Gaming tokens, which are associated with electronic gaming. We do not manually class any category.
>
> We reviewed the tags for all ERC20 tokens launched before March 2024 across these three platforms, encompassing a total of 24,316 labeled tokens.
>
> Reference
> [1] Etherscan. https://etherscan.io/.
> [2] Polygonscan. https://polygonscan.com/.
> [3] Bscscan. https://bscscan.com/.
>
>
> **Q5: Explain the pros and cons of taking a snapshot of the state of blockchain**
>
> A5: Thank you for your feedback. We agree that detailing these aspects enhances the comprehensiveness of our work. We would add this discussion to the discussion and limitation in our updated paper.
>
> Utilizing snapshots provides several advantages. First, this approach ensures data consistency, as all analyses are derived from the same state of the blockchain, thereby avoiding complications that arise from real-time data modifications. Second, processing static snapshots is computationally less demanding compared to querying live blockchain data, which enhances the feasibility of conducting large-scale analyses on complex networks. Third, the reproducibility of experiments, as the exact state of the data can be preserved and revisited for future studies.
>
> However, there are some drawbacks to this approach. First, snapshots do not capture real-time changes, which may omit transient behaviors that are crucial for certain dynamic analyses. This can potentially lead to a gap in capturing rapidly evolving trends within the blockchain ecosystem. Second, storing snapshots, especially for large and active blockchains, demands substantial storage resources, which can escalate the costs and complexity of the data management infrastructure.

---

> ### Author Response · Authors · 2024-08-23
>
> Dear Reviewer kLuX,
>
> We sincerely appreciate the time and effort you’ve devoted in reviewing our paper and offering valuable feedback. We have carefully addressed your concerns and would be grateful if you could kindly review our rebuttal. If any points need further clarification or elaboration, we would be happy to provide it. We would greatly appreciate your prompt response.
>
> Thank you once again for your time and thoughtful consideration.
>
> Best regards,
>
> Authors

---

> ### Author Response · Authors · 2024-08-25
>
> Dear Reviewer kLuX,
>
> Thank you for your valuable and detailed comments on our work, which have greatly contributed to improving the quality of our paper. As the discussion period is drawing to a close, we kindly ask if you could provide a response to our rebuttal. If any points remain unclear or if further clarification is needed, we would be more than happy to clarify. Thank you again for your time and consideration.
>
> Best regards,
>
> The Authors

---

> ### Author Response · Authors · 2024-08-28
> **Gentle Reminder to Reviewer kLuX**
>
> Dear Reviewer kLuX,
>
> We sincerely appreciate the time and effort you've dedicated to reviewing our paper. As the discussion period concluding in a few days, could you please provide a response to our rebuttal? If there are any aspects that are unclear or if you would like further explanation, we are more than happy to clarify. Thank you for your time and consideration!
>
> Best regards,
>
> The Authors

---

> ### Author Response · Authors · 2024-08-29
>
> Dear Reviewer kLuX,
>
> Thanks a lot for your time in reviewing and insightful comments. We sincerely understand you’re busy. But since the discussion due is approaching, we would really appreciate if you could check the response to confirm where you may have any further questions. We have summarized an overview of responses for your main questions for convenient reading.
>
> **Q1: Corrections to Misprints and Figure Adjustments**:
>
> A1: We added two missing references ([61] and [62]) related to outlier detection tools. Figure 4 is modified to be black and white friendly, addressing accessibility for visual representation, as shown in the attached file.
>
> **Q2: Importance of Blockchain Data and GoG**:
>
>   Blockchain data is complex and richly structured, offering unique challenges and opportunities for machine learning. It encompasses a variety of interconnected digital tokens, yet many studies overlook these interconnections, treating tokens either collectively or in isolation. Applying GoG to blockchain allows for a deeper analysis of these multi-level interactions and provides a robust framework for understanding how individual transactions influence token behaviors across different platforms.
>
> Unlike typical GoG systems in chemistry or information retrieval, which often feature smaller static graphs, blockchain GoGs consist of large-scale, dynamic graphs with dense global structures and temporal edges. This scale and dynamism pose significant computational demands and highlight the limitations of existing GoG models, especially as they struggle with complex classification tasks and anomaly detection in blockchain environments.
>
> These distinctions not only underscore the challenges but also highlight potential areas for further research and development in blockchain and graph analysis, promising to advance both fields significantly.
>
> **Q3: Experimental Design for Token-Token Interactions**:
>
> A3: We conducted experiments on predict the token-token interactions for new launched tokens use the information of the existing ones, comparing GNN models and GoG models. We noted that GoG models often did not outperform individual GNN models in predicting token interactions. This observation suggests potential areas for improvement in model design and feature engineering.
>
> **Q4: Clarity on how labels are derived, and what each category stands for**:
>
> A4: We source labels from three leading blockchain explorers—Etherscan, Polygonscan, and Bscscan—known for their reliability within the blockchain community and have been regarded as reliable by many previous blockchain studies [1, 2].
>
> For fraud cases, we labelled the tokens which are described to be suspicious phish/hack tokens by these explorers. These tokens vary, from those spammed to numerous users to those mimicking well-known tokens like fake USDT. It is worth noting that when blockchain explorers label a token, they provide not only the label but also valid justifications. For instance, token 0xbd6323a83b613f668687014e8a5852079494fb68 is labeled as "Phish/Hack" on Ethereum, with an explanation stating, “We have received reports that this is not an official token issued by BlackRock and is not associated with the brand. Please treat it with caution.” This token falsely presents itself as BlackRockTradingCurrency. We believe these highlights will help enhance the reliability of the labels. In total, 7,978 tokens were identified as suspicious, representing about 32.8% of the dataset.
>
> For other classes, we labelled the tokens using the category tags given by the explorer. For example, for token 0xa808b22ffd2c472ad1278088f16d4010e6a54d5f on Ethereum, it has tag "finance" on Etherscan which indicate it's a token related to finance. The most popular classes in the dataset include DeFi, MEME, Gaming tokens.
>
> Reference:
>
> [1] Yuan, Q., Huang, B., Zhang, J., Wu, J., Zhang, H., & Zhang, X. (2020, October). Detecting phishing scams on ethereum based on transaction records. In 2020 IEEE international symposium on circuits and systems (ISCAS) (pp. 1-5). IEEE.
>
> [2] Li, S., Gou, G., Liu, C., Hou, C., Li, Z., & Xiong, G. (2022, April). TTAGN: Temporal transaction aggregation graph network for ethereum phishing scams detection. In Proceedings of the ACM Web Conference 2022 (pp. 661-669).

---

> > ### Author Response · Authors · 2024-08-29
> >
> > **Q5:Pros and Cons of Using Blockchain Snapshots**:
> >
> > **Pros**: Snapshots provide several benefits for blockchain analysis. First, they ensure data consistency by basing all analyses on the same state, thus avoiding issues from real-time data modifications. Second, they allow for greater computational efficiency as processing static snapshots demands less computing power compared to querying live data. This facilitates more feasible large-scale analyses. Third, the use of snapshots enhances the reproducibility of experiments, allowing researchers to revisit the exact state of data for consistent replication.
> >
> > **Cons**: However, there are drawbacks to using snapshots. They do not capture real-time changes, missing transient behaviors and rapid evolutions within the blockchain, which can be crucial for dynamic analyses. Additionally, storing snapshots, especially for large and active blockchains, requires substantial storage resources. This can significantly increase the costs and complexity of managing data infrastructure.
> >
> > In addition, with the timestamps on the transaction edges in the local graphs in our dataset, global edges in our graph can also dynamically adapt to changes in the local transaction information. As detailed in Appendix D, we analyze the yearly temporal evolution of the global graph to understand how blockchain networks adapt over time. For example, for the past three years, on average, the increase of number of nodes on global graphs reaches 42.49% for Ethereum, 33.08% for Polygon, and 65.18% for BnB, indicating significant changes in the dataset.
> >
> >
> > We hope this can address your concerns. If any aspects are unclear or if you need further explanation, we are more than happy to provide clarification. Thank you very much for your time and consideration.

---

> ### Author Response · Authors · 2024-08-31
>
> Dear Reviewer kLuX,
>
> Thank you for the time and effort you've invested in reviewing our work and providing insightful feedback. We understand your busy schedule, but with the discussion deadline approaching today, could you please review our response to see if you have any further questions?
>
> We look forward to your reply and are happy to address any additional concerns.
>
> Best regards,
>
> The Authors

---

### Official Review · Reviewer_Rsox · 2024-07-26

**Rating:** 6
**Confidence:** 3
**Correctness:** The claims are mostly correct.
**Clarity:** The writing is good, but a few points…

**Review:**

Please see below for the detailed review.

**Strengths:**

1. The paper is written well and easy to understand.
2. The new dataset can be a valuable benchmark for evaluating graph neural networks, especially for graph-level tasks.
3. The authors conducted various experiments on the new dataset to show the usefulness of it.

**Additional Feedback:**

1. Do you create global edges every time when the Jaccard similarity is above 0?
2. Can you elaborate on how you determine the tags (or labels) of graphs using the blockchain explorers?
3. In Sec 5.1, the problem definition is not clear. What is the exact label to classify each local graph?
4. How exactly are the anomalies defined in Sec 5.2? Are they pre-determined tags of local graphs?

**Documentation:**

A URL is given, but no license plan is provided. Sufficient details are given in the text.

**Ethics:**

As the authors say, I believe there might be a privacy issue, since one may recognize people’s identities by combining all the transactions existing in the data. The chance will be unlikely though.

**Limitations:**

The authors reveal the limitations of this work.

**Opportunities For Improvement:**

1. I’m not sure how much meaningful the global edges are, since they are manually created by the authors. Are there any previous works that create edges between graphs in a similar way?
2. A few details are unclear from the text. Please see the additional feedback below.
3. The conclusions from experiments are not justified well. For example, in Line 225, the authors claim that “GoG models exhibit superior performance compared to individual GNN models across most tasks in both classification scenarios,” but the difference does not seem to be significant from the table. At the same time, considering that there are multiple models for each category, a formal statistical analysis is required to have such a conclusion.

**Relation To Prior Work:**

The difference from existing datasets is well discussed.

**Summary And Contributions:**

This paper introduces a novel dataset that provides label information at the token level and integrates token-token interactions on multiple blockchain platforms. Specifically, the authors model transactions of each token as local graphs and the relationships between tokens as global graphs, collectively forming a Graphs of Graphs (GoG) system.

---

> ### Author Rebuttal · Authors · 2024-08-17
>
> **Q1: I’m not sure how much meaningful the global edges are, since they are manually created by the authors. Are there any previous works that create edges between graphs in a similar way?**
>
> A1: Thank you for your feedback. In our paper, the global graphs model the correlation of various tokens across blockchains. In Section 4.2, we provide an analysis demonstrating that global edges are instrumental in identifying tokens within the same class and revealing strong relationships between transaction activity and centrality within the blockchain ecosystem. There are previous works create edges between graphs in a similar way. For example, a GoG study on multiple social media groups that constructed a global graph through common members across these groups [1]. Another study on citation networks from different research areas that built global graph edges by measuring the ratio of citations between these areas to the total number of cited papers [2]. Similarly, in our analysis, global edges reveal significant correlations between token transactions across platforms, offering deeper insights into market behavior.
>
> [1] Li, J., Rong, Y., Cheng, H., Meng, H., Huang, W., & Huang, J. (2019, May). Semi-supervised graph classification: A hierarchical graph perspective. In The World Wide Web Conference (pp. 972-982).
>
> [2] Ni, J., Tong, H., Fan, W., & Zhang, X. (2014, August). Inside the atoms: ranking on a network of networks. In Proceedings of the 20th ACM SIGKDD international conference on Knowledge discovery and data mining (pp. 1356-1365).
>
>
>
>
> **Q2: A few details are unclear from the text. Please see the additional feedback below.**
>
> **Q2.1: Do you create global edges every time when the Jaccard similarity is above 0?**
>
> A2.1: Thank you for your question. We use Jaccard similarity as the weight of the global edges. In our experiments in Section 5, we applied a threshold for edge weights, only considering connections in the global graph where the weight surpassed 0.01. This criterion was established to focus on significantly interconnected tokens, thereby emphasizing more substantial relationships.
>
> **Q2.2, Q2.3, and Q2.4: Can you elaborate on how you determine the tags (or labels) of graphs using the blockchain explorers?
> In Sec 5.1, the problem definition is not clear. What is the exact label to classify each local graph?
> How exactly are the anomalies defined in Sec 5.2? Are they pre-determined tags of local graphs?**
>
>
> A2.2/3/4: Thank you for your valuable feedback. Thank you for your valuable feedback. We recognize the importance of transparency in data source and labeling for maintaining the quality and trustworthiness of research in blockchain research.
>
> In Section 3.1 of our paper, we initially described our collection of labels sourced from three leading blockchain explorers: Etherscan [1], Polygonscan [2], and Bscscan [3]. These platforms are renowned for their extensive use and reliable data within the blockchain community. In response to your concerns, we have enhanced our description to include our systematic process for label verification and class grouping, which we detail below (this part is to be added to Section 3.1):
>
> For fraud cases, we labelled the tokens which are described to be suspicious phish/hack tokens by these explorer. Those tokens include various kinds of spammed tokens, such as those have been spammed to many users or those who pretend to be some famous tokens, such as fake USDT. For example, for token 0x8f1a7720d0798d50a01111b15f8c48d43cbc96af on Ethereum, it is highlighted as “This token is reported to have been spammed to many users. Please exercise caution when interacting with it.” on its token page on Etherscan [https://etherscan.io/token/0x8f1a7720d0798d50a01111b15f8c48d43cbc96af]. In total, 7,978 tokens were identified as suspicious, representing about 32.8% of the dataset.
>
> For other classes, we labelled the tokens using the category tags given by the explorer. For example, for token 0xa808b22ffd2c472ad1278088f16d4010e6a54d5f on Ethereum, it has tag "finance" on Etherscan which indicate it's a token related to finance.[https://etherscan.io/token/0xa808b22ffd2c472ad1278088f16d4010e6a54d5f]. The most popular classes in the dataset include DeFi tokens, which are related to decentralized finance products; MEME tokens, often inspired by internet memes and characters; and Gaming tokens, which are associated with electronic gaming. We do not manually class any category.
>
> We reviewed the tags for all ERC20 tokens launched before March 2024 across these three platforms, encompassing a total of 24,316 labeled tokens.
>
> Reference
> [1] Etherscan. https://etherscan.io/.
> [2] Polygonscan. https://polygonscan.com/.
> [3] Bscscan. https://bscscan.com/.

---

> ### Author Rebuttal · Authors · 2024-08-17
>
> **Q3: The conclusions from experiments are not justified well. For example, in Line 225, the authors claim that “GoG models exhibit superior performance compared to individual GNN models across most tasks in both classification scenarios,” but the difference does not seem to be significant from the table. At the same time, considering that there are multiple models for each category, a formal statistical analysis is required to have such a conclusion.**
>
> A3: Thank you for your feedback and suggestion. From Table 4, we can observe that in most experiments, except for the 5-class classification of the Polygon dataset, GoG models, either SEAL or GoGNN, show better performance compared to all other individual GNN models across most tasks. However, DVGGA as another GoG model seem to show worse performance.
>
> In addition, we have conducted statistical analysese to verify the significance of these results as below:
>
> (1)**percentage increase calculation**: we compare the average performance of individual models (e.g., GCN, GAT, etc.) against GoG models (i.e., SEAL, GoGNN) by calculating the relative percentage improvement in their performance. This gives a straightforward assessment of the performance of the GoG models.
>
> (2) **paired t-test**: we compare the paired F1-macro and F1-micro scores for GoG models against individual models to determine if there is a significant difference in performance across datasets. A low p-value (typically below 0.05) indicates that the differences in performance are statistically significant.
>
> The results are presented in tables of the attached PDF. As shown, SEAL consistently outperforms individual models in the BNB dataset, with statistically significant improvements in both F1-macro (24.25%, p < 0.05 for 3-class; 19.31%, p < 0.05 for 5-class) and F1-micro (12.69%, p < 0.05 for 3-class; 7.70%, p < 0.05 for 5-class). Additionally, SEAL significantly improves performance in Ethereum for the 5-class task (36.46%, p < 0.05 for F1-macro; 6.08%, p < 0.05 for F1-micro). In Polygon, GoGNN shows a significant improvement in F1-macro (22.20%, p < 0.05 for 3-class; 12.79%, p < 0.05 for 5-class), but exhibits a decline in F1-micro.

---

> > ### Comment · Reviewer_Rsox · 2024-08-19
> >
> > Thank you for the response. I will keep my positive score.

---

> > > ### Author Response · Authors · 2024-08-23
> > >
> > > Dear Reviewer Rsox,
> > >
> > > Thank you once again for your valuable input and consideration!
> > >
> > > Best regards,
> > > Authors

---

### Official Review · Reviewer_RMtu · 2024-07-31
**A graph of graphs dataset for blockchains**

**Rating:** 7
**Confidence:** 5
**Correctness:** Yes, the claims are correct
**Clarity:** The article is well written

**Review:**

The paper has two main objectives. The first is to compare the graph characteristics of Ethereum-based blockchains like Polygon, BNB, and Ethereum. The authors achieve this by providing statistics on the graph properties of each network, offering a straightforward comparison of their behaviors.

The second objective is to apply a Graphs of Graphs framework to these networks. I really like this approach. Using this framework, the authors develop supervised machine learning models that classify graphs and predict anomalies within these blockchains. This approach represents an extension of traditional graph analysis to incorporate a more complex hierarchical model that captures the intricate relationships in blockchain data.

**Strengths:**

- This paper is among the few that compare multiple blockchain data within a single study, building on precedents like the Chartalist paper. Unlike Chartalist, which does not provide detailed graph statistics, this paper offers a comprehensive comparison of graph characteristics across Polygon, BNB, and Ethereum. This comparison helps us understand how these networks vary in terms of their graph properties, providing insights for blockchain analysis.
- The application of the GoG framework to blockchain networks is a novel approach that opens up new research possibilities in the field. This method is particularly promising as it potentially reduces the reliance on labeled data, leveraging similarities within the data instead.

**Additional Feedback:**

The article offers a lot, but it is not yet ready to be published. I believe that a website with loader libraries must be developed (as in Chartalist) and resubmitted next year (as Chartalist had done).

**Documentation:**

The documentation is missing. The dataset should have been given on a well explained URL with loaders etc.

**Ethics:**

I do not see an ethics issue

**Limitations:**

yes, however an address exclusion form must have been created for the data.

**Opportunities For Improvement:**

- A major concern with the paper is the lack of information on where the labeled data came from. It's not clear how the authors decided which cases were fraud or how they grouped the data into different classes. This lack of detail about how the data was labeled is a big issue. For a conference like NeurIPS, it's important that all research clearly shows where their data comes from to avoid any bias that might happen if the data is labeled incorrectly or unfairly. It’s crucial for maintaining the quality and trustworthiness of the research, especially in a field as important as blockchain technology (hoping we see more article in this field).
- The authors must detail exclusion results better, and make sure/prove that data exclusion does not impact the results. I believe that the rigor in answering this review will play a major part in this article's future.
- The models aspect can be improved, because the tasks are not that interesting. The GoG approach can be used to frame investment prediction (will this token attrack the investors of that token?) as a future edge prediction, hence models such as HTGN (no relation to me)  can be run on the GoG.

HTGN: Menglin Yang, Min Zhou, Marcus Kalander, Zengfeng Huang, and Irwin King. Discrete-time temporal network embedding via implicit hierarchical learning in hyperbolic space. In Proceedings
of the 27th ACM SIGKDD Conference on Knowledge Discovery & Data Mining, pp. 1975–1985,
2021

**Relation To Prior Work:**

Yes, blockchain repositories and datasets are especially detailed.

**Summary And Contributions:**

The paper introduces a new dataset for machine learning on blockchain graphs, focusing on a model called Graphs of Graphs. This framework represents blockchain tokens as local graphs for individual transactions and global graphs for interactions between tokens. It includes data from multiple EVM platforms and provides detailed graph-level analysis capabilities, which have been limited in previous studies. The research explores token classifications and anomaly detection through machine learning, building predictive models for blockchain dynamics and token interactions.

---

> ### Author Rebuttal · Authors · 2024-08-17
>
> **Q1: A major concern with the paper is the lack of info...**
>
> A1: Thank you for your valuable feedback. We recognize the importance of transparency in data source and labeling for maintaining the quality and trustworthiness of research in blockchain research.
>
> In Section 3.1 of our paper, we initially described our collection of labels sourced from three leading blockchain explorers: Etherscan [1], Polygonscan [2], and Bscscan [3]. These platforms are renowned for their extensive use and reliable data within the blockchain community. In response to your concerns, we have enhanced our description to include our systematic process for label verification and class grouping, which we detail below (this part is to be added to Section 3.1):
>
> For fraud cases, we labelled the tokens which are described to be suspicious phish/hack tokens by these explorer. Those tokens include various kinds of spammed tokens, such as those have been spammed to many users or those who pretend to be some famous tokens, such as fake USDT. For example, for token 0x8f1a7720d0798d50a01111b15f8c48d43cbc96af on Ethereum, it is highlighted as “This token is reported to have been spammed to many users. Please exercise caution when interacting with it.” on its token page on Etherscan [https://etherscan.io/token/0x8f1a7720d0798d50a01111b15f8c48d43cbc96af]. In total, 7,978 tokens were identified as suspicious, representing about 32.8% of the dataset.
>
> For other classes, we labelled the tokens using the category tags given by the explorer. For example, for token 0xa808b22ffd2c472ad1278088f16d4010e6a54d5f on Ethereum, it has tag "finance" on Etherscan which indicate it's a token related to finance.[https://etherscan.io/token/0xa808b22ffd2c472ad1278088f16d4010e6a54d5f]. The most popular classes in the dataset include DeFi tokens, which are related to decentralized finance products; MEME tokens, often inspired by internet memes and characters; and Gaming tokens, which are associated with electronic gaming. We do not manually class any category.
>
> We reviewed the tags for all ERC20 tokens launched before March 2024 across these three platforms, encompassing a total of 24,316 labeled tokens.
>
> Reference:
> [1] Etherscan. https://etherscan.io/.
> [2] Polygonscan. https://polygonscan.com/.
> [3] Bscscan. https://bscscan.com/.
>
>
> **Q2: The authors must detail exclusion...**
>
> A2: Thank you for your constructive feedback. We understand the importance of rigorously detailing and justifying any data exclusions in our experiments to ensure the integrity of our results. We implemented specific token graphs filtering to enhance the reliability and focus of our experiments. This is because small graphs lack sufficient transactional data to contribute meaningfully to the model.
>
> In specific, we excluded token graphs with fewer than five nodes or edges. This threshold ensures the statistical significance and stability of the network metrics calculated, as graphs smaller than this often lack sufficient data to provide meaningful insights or validate network theories reliably. After applying this criterion, only less than 2% of tokens were removed from all three datasets. This minimal exclusion ensures that our analysis still represents the vast majority of the data, maintaining the robustness and generalizability of our findings.
>
>
>
> **Q3: The models aspect can be improved...**
>
> A3: Thank you for your suggestion regarding the use of the HTGN model. We have carefully reviewed it. As a novel hyperbolic temporal graph embedding model, it excels at learning temporal regularities, topological dependencies, and implicitly hierarchical organization. While it models the global graph effectively, it is not specifically designed to capture the information within local graphs, making it less directly applicable to our GoG setting.
>
> To address this and explore the potential of HTGN, we adapted our approach to apply HTGN to the global graph component of our dataset. For this experiment, we focused on the most current tokens that launched in the last year, and we configured the time granuality of the temporal data to be monthly. This adaptation allows us to still benefit from HTGN's strengths in handling temporal and hierarchical data. Following the settings in the HTGN paper, we evaluated the models on two dynamic link prediction tasks: temporal link prediction and temporal new link prediction. Specifically, given partially observed snapshots of a temporal graph \( G = \{G_1, ..., G_t\} \), the dynamic link prediction task involves predicting links in the next snapshot \( G_{t+1} \) or in future snapshots over multiple steps. The dynamic new link prediction task focuses on predicting all new links in \( G_{t+1} \) that did not exist in \( G_t \).
>
> The results of these experiments are shown in the Table in the Appendix. From the results, we can see that HTGN achieves high AUC and AP scores on these three datasets for temporal link prediction, all exceeding 93%. However, the performance for temporal new link prediction is relatively weak, especially for Ethereum, where it falls below 70%. This suggests that while HTGN effectively captures existing patterns and interactions within the datasets, it may be less adept at predicting emerging connections in dynamically evolving blockchain environments. This finding highlights potential areas for further enhancement in modeling techniques to better predict new links as token interactions evolve.
>
>
> **Documentation: The documentation is missing. The dataset should have been given...**
>
> A: Thank you for your reminder. Our code is documented in GitHub [https://anonymous.4open.science/r/Graph-of-graph-dataset-B2E3/README.md] as stated in Appendix I. In the repo, we have included instructions to ensure that users can quickly start working with the dataset without extensive setup. We would move it to the paper abstract.

---

> ### Author Response · Authors · 2024-08-23
>
> Dear Reviewer RMtu,
>
> We sincerely appreciate the time and effort you have dedicated to reviewing our paper and providing valuable feedback. We have worked diligently to address your concerns.
>
> We kindly request that you take a moment to review our rebuttal. Should any points remain unclear or if further elaboration is needed, we are more than happy to provide additional clarification. We would appreciate your prompt response.
>
> Thank you again for your time and consideration.
>
> Best regards,
>
> Authors

---

> ### Comment · Reviewer_RMtu · 2024-08-23
> **Thanks**
>
> The exclusion criteria was not as bad as i expected, and the labeling comes from the websites that we also use. As a result my concerns have been addressed. Thank you for running htgn on the dataset as well.
>
> Code should be given in the main article.
>
> I will increase me rating to 7.

---

> > ### Author Response · Authors · 2024-08-23
> >
> > Dear Reviewer RMtu,
> >
> > Thank you for your constructive feedback! We’re glad we could address your concerns and sincerely appreciate your decision to increase the rating. We will incorporate your suggestions into the paper. Your input has been invaluable.
> >
> > Thank you again for your support!
> >
> > Best regards,
> >
> > Authors

---

### Decision · Program_Chairs · 2024-09-26

**Decision:**

Accept (Poster)

**Comment:**

The paper presents a new blockchain dataset, where blockchain transactions are organized as a graph-of-graphs. All the reviewers agree that the application domain is somewhat novel, the process of dataset curation is sound, and the benchmarking experiments are appropriate. The dataset could also be very useful to the graph-based machine learning community. The reviewers raised some concerns about the labeling process, exclusion criteria, nature of the proposed tasks, and claims about the benchmarking results. In my opinion, the authors have addressed all these concerns satisfactorily through their responses. Hence, I recommend acceptance of the paper.